# ULK3 regulates cytokinetic abscission by phosphorylating ESCRT-III proteins

Anna Caballe[1†], Dawn M Wenzel[2†], Monica Agromayor[1], Steven L Alam[2], Jack J Skalicky[2], Magdalena Kloc[1], Jeremy G Carlton[1‡], Leticia Labrador[1], Wesley I Sundquist[2]*, Juan Martin-Serrano[1]*

[1]Department of Infectious Diseases, King's College London School of Medicine, London, United Kingdom; [2]Department of Biochemistry, University of Utah School of Medicine, Salt Lake City, United States

**Abstract** The endosomal sorting complexes required for transport (ESCRT) machinery mediates the physical separation between daughter cells during cytokinetic abscission. This process is regulated by the abscission checkpoint, a genome protection mechanism that relies on Aurora B and the ESCRT-III subunit CHMP4C to delay abscission in response to chromosome missegregation. In this study, we show that Unc-51-like kinase 3 (ULK3) phosphorylates and binds ESCRT-III subunits via tandem MIT domains, and thereby, delays abscission in response to lagging chromosomes, nuclear pore defects, and tension forces at the midbody. Our structural and biochemical studies reveal an unusually tight interaction between ULK3 and IST1, an ESCRT-III subunit required for abscission. We also demonstrate that IST1 phosphorylation by ULK3 is an essential signal required to sustain the abscission checkpoint and that ULK3 and CHMP4C are functionally linked components of the timer that controls abscission in multiple physiological situations.

*For correspondence: wes@ biochem.utah.edu (WIS); juan. martin_serrano@kcl.ac.uk (JM-S)

†These authors contributed equally to this work

Present address: ‡Division of Cancer Studies, King's College London School of Medicine, London, United Kingdom

## Introduction

Cytokinesis is the final stage of cell division when two daughter cells are physically separated. The process comprises several steps, including cleavage furrow ingression and abscission of the midbody—the thin intercellular bridge connecting the nascent daughter cells (*Caballe and Martin-Serrano, 2011*; *Mierzwa and Gerlich, 2014*). Midbody abscission occurs adjacent to the Flemming body, a central protein complex, and is mediated by the ESCRT pathway (endosomal sorting complexes required for transport) (*Carlton and Martin-Serrano, 2007*; *Morita et al., 2007*). The ESCRT machinery performs topologically equivalent membrane fission events throughout the cell, including intraluminal vesicle formation at the multivesicular body, retroviral particle release, plasma membrane wound repair and abscission (*Hurley and Yang, 2008*; *Henne et al., 2011*; *McCullough et al., 2013*; *Jimenez et al., 2014*). The core ESCRT machinery comprises five different factors or complexes: ALIX, ESCRT-I, ESCRT-II and ESCRT–III, and VPS4. In most cases, location-specific adaptors initially recruit the early-acting ALIX and ESCRT-I/-II factors, which then recruit the late-acting ESCRT-III and VPS4 complexes to mediate membrane fission. In late stages of cytokinesis, the Flemming body protein CEP55 binds TSG101 (ESCRT-I) and ALIX, which leads to ESCRT-III subunit recruitment (*Carlton and Martin-Serrano, 2007*; *Morita et al., 2007*; *Carlton et al., 2008*). ESCRT-III-based helical filaments form on either side of the Flemming body and induce cortical constriction, thereby promoting abscission (*Elia et al., 2011*; *Guizetti et al., 2011*). In addition to driving membrane scission mechanically, ESCRT-III proteins recruit cytokinetic cofactors that contain MIT (microtubule interacting and trafficking) domains. MIT domains bind short motifs within the C-terminal tails of ESCRT-III proteins called MIMs (MIT-interacting motifs), which become exposed and

**eLife digest** Our cells multiply by dividing into two. Many proteins are involved in this process, including a group called the ESCRT-III complex. This group is required to complete the final stage of cell division when the single membrane that surrounds the two new daughter cells separates. Before the cell divides, its DNA—which is packaged in structures called chromosomes—is copied, and the two sets of chromosomes are pulled to opposite ends of the cell. This ensures that each daughter cell will have a complete set of DNA.

If the cell divides before the chromosomes have finished moving to opposite ends of the cell, the daughter cells may end up with the wrong number of chromosomes. This can lead to cancer or other diseases. To prevent this, cells have evolved a quality control system called the 'abscission checkpoint', which delays cell division until the chromosomes have properly separated. Previous studies have shown that when the checkpoint is active, an ESCRT-III complex protein called CHMP4C is inactivated by an enzyme, which prevents the cell from dividing.

Other signals that indicate that the new daughter cells are not yet ready to separate can also delay cell division, but it is not clear how those defects are detected by the checkpoint. Here, Caballe, Wenzel et al. found that a protein called ULK3 can bind to several proteins in the ESCRT-III complex, including one called IST1. In doing so, ULK3 is able to delay cell division if the chromosomes have not finished separating, if there are defects in the nucleus of the cell, or if the cell is experiencing high levels of mechanical tension at the site where the membrane will separate.

The experiments also show that ULK3 needs to bind to and regulate the activity of IST1 to sustain the abscission checkpoint, and that CHMP4C is required for this process. Caballe, Wenzel et al.'s findings reveal that ULK3 plays an essential role in controlling when a cell divides and imply that there may be additional proteins involved that release cells from the checkpoint delay imposed by ULK3. The next challenges will be to identify these proteins and to understand how all checkpoint proteins work together to regulate cell division.

concentrated when the ESCRT-III proteins polymerize (*McCullough et al., 2013*). MIT domain-containing proteins that participate in cytokinesis include the AAA-ATPase VPS4 and its activator LIP5, which remodel and recycle the ESCRT-III subunits (*Babst et al., 1998*; *Hurley and Yang, 2008*; *Lata et al., 2008*), Spastin, an AAA-ATPase that severs microtubules, and MITD1, a membrane-binding protein that stabilizes the intercellular bridge (*Yang et al., 2008*; *Connell et al., 2009*; *Skalicky et al., 2012*; *Hadders et al., 2012*; *Azmi et al., 2006*).

Evolutionarily conserved monitoring mechanisms regulate the proper timing of events during cytokinesis progression (*Agromayor and Martin-Serrano, 2013*; *Mierzwa and Gerlich, 2014*). When chromosomes persist within the midbody, the abscission checkpoint (also known as NoCut) inhibits abscission until chromatin has been cleared from the spindle midzone, thereby preventing aberrant segregation or cleavage furrow regression and tetraploidy (*Norden et al., 2006*; *Mendoza et al., 2009*; *Steigemann et al., 2009*). Lagging chromosomes are sensed by the Aurora B kinase, which phosphorylates the ESCRT-III subunit CHMP4C to delay abscission (*Capalbo et al., 2012*; *Carlton et al., 2012*). Defective nuclear pore complex assembly also triggers Aurora B-dependent abscission delays (*Mackay et al., 2010*). The full signaling cascade that connects nuclear pores to abscission is not yet clear, but ESCRT proteins themselves are involved in the surveillance and clearance of defective nuclear pore complex assembly intermediates in *S. cerevisiae* (*Webster et al., 2014*). Tension forces applied by dividing cells on the midbody also regulate cytokinesis, with high-tension delaying abscission, and tension release triggering ESCRT-III assembly and membrane scission (*Lafaurie-Janvore et al., 2013*). How these different physiological inputs converge to influence abscission timing is not understood.

Here, we investigate the function of Unc-51-like kinase 3 (ULK3), a poorly characterized member of the ULK family of serine/threonine kinases that is predicted to contain tandem MIT domains (*Row et al., 2007*). Live-cell imaging analysis revealed that ULK3 regulates abscission timing in response to lagging chromosomes, defects in nuclear pore complex assembly, and tension forces at the midbody. Furthermore, our biochemical and structural studies show that the ULK3 MIT domains bind tightly to IST1, an ESCRT-III subunit required for cytokinesis (*Agromayor et al., 2009*; *Bajorek et al., 2009a*).

Finally, we show that ULK3 phosphorylates IST1 and other ESCRT-III proteins and that IST1 phosphorylation provides an essential inhibitory signal in the abscission checkpoint, thereby ensuring proper coordination of the final events in cell division.

## Results

### ULK3 binds to ESCRT-III via tandem MIT domains

The predicted MIT domains in ULK3 suggested a novel mechanism of ESCRT regulation, and we, therefore, surveyed potential ULK3–ESCRT interactions using yeast two-hybrid (Y2H) experiments. ULK3 binding was observed for three ESCRT-III subunits: CHMP1A, CHMP1B, and CHMP2A, but not for other ESCRT complexes (*Figure 1—figure supplement 1A*). These interactions were confirmed by co-immunoprecipitation of Myc-tagged ESCRT-III proteins from mixed 293T cell lysates that contained One-strep-flag (OSF)-tagged ULK3 (*Figure 1A*, note interactions in lanes 2, 4, 6, and 14). This approach revealed that ULK3 also bound the ESCRT-III subunit IST1 (lane 26), an interaction not tested by Y2H because IST1 fusion constructs activated transcription non-specifically. Endogenous ULK3 also co-precipitated with overexpressed HA-tagged CHMP1A, CHMP1B, CHMP2A, or IST1, but not with CHMP2B (*Figure 1B*). Finally, endogenous IST1 was efficiently biotinylated in cells that expressed a biotin ligase BirA-ULK3 fusion protein, which promiscuously biotinylates proximal proteins (*Figure 1C*, lane 4, bottom panel) (*Roux et al., 2012*). Hence, ULK3 can interact with a specific subset of ESCRT-III proteins in cells.

### ULK3 MIT2 binds IST1 MIM1

Fluorescence polarization (FP) binding assays indicated that ULK3 interacted more tightly with IST1 ($K_D \sim 0.2$ μM) than with other ESCRT-III proteins ($K_D \sim 100$ μM) (*Figure 1F*, and data not shown). We, therefore, focused on characterizing the IST1-ULK3 interaction. The IST1-binding site on ULK3 was mapped using co-precipitation experiments with Myc-tagged IST1 and different OSF-ULK3 deletion constructs. IST1 bound with similar efficiencies to full-length ULK3 (residues 1–472) and to a minimal construct that spanned just the C-terminal tandem MIT domains (residues 277–449, hereafter denoted ULK3(MIT)$_2$; *Figure 1D*, lanes 2 and 5, respectively), but did not bind to the isolated ULK3 kinase domain (residues 1–270; lane 3). Thus, the ULK3 MIT domains form the primary IST1-binding site. As discussed below, co-expression of full-length ULK3 resulted in the appearance of lower mobility IST1 species, suggesting that the kinase can phosphorylate IST1 (*Figure 1D*, lane 2).

   NMR chemical shift perturbation experiments were used to map ULK3-binding sites on IST1. IST1 contains two different MIT interaction motifs, termed MIM1 (residues 352–363) and MIM2 (residues 325–329), that were included within a C-terminal IST1 peptide used in these experiments (IST1$_{303-366}$) (*Agromayor et al., 2009*; *Bajorek et al., 2009a*). Both MIMs exhibited backbone amide proton chemical shift perturbations upon titration of $^{15}$N-labeled IST1$_{303-366}$ with ULK3(MIT)$_2$, indicating that both IST1 MIMs can contact ULK3(MIT)$_2$ (*Figure 1E*), whereas the upstream IST1 residues 303–317 were not perturbed. To assess the binding contributions of each MIM and MIT element, we performed FP-binding experiments with fluorescently labeled peptides comprising either one or both of the IST1 MIM elements and one or both ULK3 MIT domains. The tightest interaction was observed for ULK3 (MIT)$_2$ binding to a construct that spanned both IST1 MIMs (*Figure 1F*). However, nearly all of the binding energy was contributed by the IST1 MIM1 element, which bound ULK3 MIT2. Consistent with these experiments, IST1$_{316-366}$ binding to ULK3(MIT)$_2$ was reduced more than 100-fold by a single point mutation in the MIM1-binding site of ULK3 MIT2 (M434D), but not by an equivalent mutation in the MIM1-binding site of ULK3 MIT1 (residues 277–358, V338D mutant; <twofold, *Figure 1F* and *Figure 1—figure supplement 1D*). Hence, both IST1 MIM elements can contact both ULK3 MIT domains, but IST1 MIM1 binding to ULK3 MIT2 domain is the energetically dominant interaction.

### Structure of IST1 MIM1-ULK3 MIT2 complex

We determined the crystal structure of ULK3 MIT2 in complex with IST1 MIM1 to 1.4 Å resolution (*Figure 1G* and *Figure 1—source data 1*). ULK3 residues 372–446 form the characteristic MIT three-helix bundle, and the IST1 MIM1 helix binds parallel to MIT helix 3 in the groove between MIT helices 2 and 3 (*Figure 1G* and *Figure 1—figure supplement 1B*). The N-terminus of the MIT domain (residues 360–373) forms a short helix that packs in the groove connecting MIT helices 1 and 3. This helix has not been seen in previous MIT domain structures and is expected to interfere sterically with

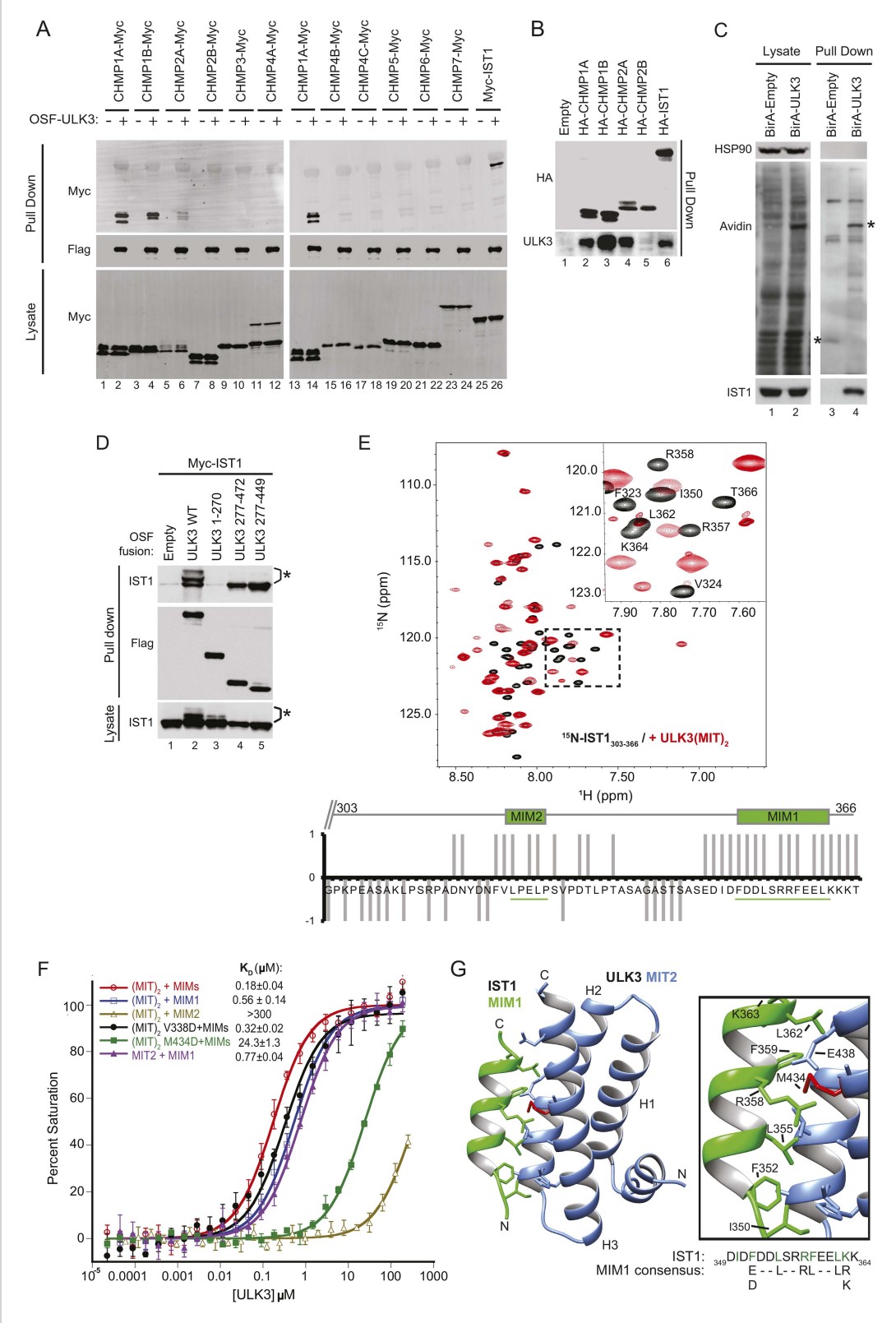

**Figure 1**. ULK3 binds ESCRT-III via tandem MIT domains. (**A**) Lysates from 293T cells overexpressing Myc-endosomal sorting complexes required for transport (ESCRT)-III proteins were mixed with lysates from cells non-transfected (–) or overexpressing One-strep-flag (OSF)-Unc-51-like kinase 3 (ULK3) (+). OSF-ULK3 proteins were bound to streptactin resin and bound ESCRT-III proteins were detected with α-Myc antibody (top). (**B**) Lysates from 293T cells

*Figure 1. continued on next page*

*Figure 1. Continued*

expressing HA-ESCRT-III were immunoprecipitated with α-HA antibody and co-precipitated endogenous ULK3 protein was detected by Western blot with α-ULK3 antibody. (**C**) HeLa cells expressing ULK3 fused to the biotin protein ligase BirA-113G or unfused BirA were treated overnight with biotin. Vicinal biotinylated proteins were isolated with streptavidin-coated beads, and endogenous IST1 was found to be biotinylated, implying that it was in close proximity with ULK3. 'Asterisks' denote isolated BirA-Empty and BirA-ULK3, respectively, on α-Avidin blot (lanes 3 and 4). Images shown for both lysate and pull-down samples were cropped from the same blot in all cases. (**D**) Co-immunoprecipitation of Myc-IST1 with different ULK3 constructs. 'Asterisks' denote phosphorylated IST1 species. (**E**) Top: overlaid $^{15}$N-HSQC NMR spectra of $^{15}$N,$^{13}$C-IST1 (residues 303–366) alone (black) or with 2 equivalents of ULK3(MIT)$_2$ (residues 277–449) (red). Inset shows an expanded view of the boxed region in the spectrum. Bottom: individual IST1 backbone amide resonances perturbed (1) or not (−1) upon addition of ULK3(MIT)$_2$. Ambiguous changes (due to spectral overlap) were scored as 0. (**F**) Fluorescence polarization (FP)-binding isotherms show interactions between different ULK3 MIT constructs corresponding to MIT2 (residues 359–449) or both MIT domains (ULK3(MIT)$_2$) and fluorescently labeled IST1 peptides for MIT-interacting motifs (MIM) 1, MIM2, or both MIM1+MIM2 (MIMs). Data points are averages ±SD of at least three separate measurements, and curves show fits to simple 1:1 binding models with dissociation constants. (**G**) Crystal structure of the ULK3 MIT2-IST1 MIM1 complex. Inset shows key interacting side-chains (in green) with ULK3 M434 highlighted in red. Sequence alignment compares IST1 MIM1 to the consensus MIM1 sequence. See also *Figure 1—figure supplement 1* and *Figure 1—source data 1*.

The following source data and figure supplement are available for figure 1:

**Source data 1**. Data Collection and Refinement Statistics for the ULK3 MIT2:IST1 MIM1 Complex.

**Figure supplement 1**. ULK3 binds to ESCRT-III via tandem MIT domains.

the binding of canonical MIM2 (and MIM3) elements (*Figure 1—figure supplement 1C*). Core IST1 MIM1 residues make a series of interactions that are similar to those seen in other MIT-MIM1 complexes (see *Figure 1F* magnification and *Figure 1—figure supplement 1B*) (*Obita et al., 2007*; *Stuchell-Brereton et al., 2007*; *Hadders et al., 2012*; *Guo and Xu, 2015*). The ULK3 M434 residue is central to this interface (*Figure 1G*, red), which explains why the M434D mutation inhibits IST1 binding (*Figure 1F* and *Figure 1—figure supplement 1D*).

The interaction between ULK3 MIT2 and IST1 MIM1 ($K_D$ = 0.77 μM) is significantly tighter than many previously described MIT–MIM1 interactions ($K_D$ ~2–100 μM). The additional binding energy appears to be contributed by N-terminal IST1 MIM1 residues I350 and F352, and by the central F359 residue, all of which make hydrophobic interactions that are absent in other MIM1-MIT complexes. Binding of the bulky hydrophobic IST1 MIM1 F352 and F359 residues is accommodated by complementary small ULK3 MIT2 interfacial residues A415 and G408, respectively. Charged or polar residues are found in equivalent positions of other MIT domains such as VPS4A (*Figure 1—figure supplement 1B*) or ULK3 MIT1. These interactions likely explain why IST1 MIM1 binds tightly and specifically to the ULK3 MIT2 domain.

ULK3 binding also mapped to the MIM1 of CHMP1B as binding was inhibited by removal of CHMP1B MIM1 or by MIM1 point mutations (*Figure 1—figure supplement 1E*, lanes 3–7 vs lane 2), but not by a mutation in the more extended CHMP1B MIM3 binding site (lane 8) (*Yang et al., 2008*). However, unlike IST1, the binding of CHMP1A and CHMP1B was reduced by point mutations in the MIM1-binding sites of either ULK3 MIT domain (*Figure 1—figure supplement 1D*, lanes 1–4 and *Figure 1—figure supplement 1F*). Thus, the binding specificities of the CHMP1A/B and IST1 proteins differ, and our data indicate that CHMP1 MIM1 elements can bind either ULK3 MIT domain.

## ULK3 regulates abscission timing and is an essential component of the abscission checkpoint pathway

We used siRNA depletion to test whether ULK3 participates in endosomal sorting or virus budding. Effects on ESCRT-dependent lysosomal targeting were tested using two established assays; down-regulation of MHC class I by the Kaposi sarcoma-associated herpesvirus (KSHV) K3 protein (*Hewitt et al., 2002*), and degradation of Tetherin by the KSHV K5 protein (*Agromayor et al., 2012*). ULK3 depletion did not affect either of these ESCRT-dependent degradation reactions, whereas control depletion of TSG101 inhibited both processes (*Figure 2—figure supplement 1A,B*). Similarly, ULK3 depletion had no effect on ESCRT-dependent HIV-1 budding (*Figure 2—figure supplement 1C*). We, therefore, conclude that ULK3 is not required for either of these ESCRT pathway functions.

We next tested whether ULK3 functions in cytokinetic abscission using live imaging of HeLa mCherry-Tubulin cells. ULK3-depleted cells proceeded through mitosis normally, but resolved their midbodies faster than control cells, at rates comparable to the abnormally rapid abscission observed in CHMP4C-depleted cells (*Figure 2A* and *Figure 2—figure supplement 1D*, *Videos 1–3*) (*Carlton et al., 2012*). In addition, endogenous ULK3 protein was detected by immunofluorescence staining at the Flemming body, where it is positioned to function during cytokinetic abscission (*Figure 2B*). Based on these results, we tested whether ULK3 was required for abscission delays induced by the abscission checkpoint. One event that triggers the abscission checkpoint is the presence of defectively assembled nuclear pore complexes (*Mackay et al., 2010*). As expected, partial depletion of Nucleoporin 153 (NUP153) induced abscission delays that increased the number of midbody-connected cells (*Figure 2C*, lane 1 vs 2). This phenotype was abrogated upon ULK3-depletion, despite comparable levels of NUP153 depletion (lane 2 vs 4). Active Aurora B (Aurora B pT232) is known to localize to the midbody of dividing cells both in control or NUP153-depleted cells (*Mackay et al., 2010*). Immunofluorescence imaging of cells treated as in *Figure 2C* revealed that ULK3 depletion did not alter the localization of phosphorylated Aurora B to the midbody in cells co-depleted of NUP153, indicating that ULK3 likely functions downstream of Aurora B in the abscission checkpoint (*Figure 2—figure supplement 1E*). These results were confirmed by live-cell imaging experiments, which demonstrated that ULK3 depletion abolished the 30-min abscission delay triggered by nuclear pore disruption (*Figure 2D*). Furthermore, partial depletion of NUP153 in two different clonal ULK3-knockout cell lines, in which the endogenous ULK3 locus was removed using a lentiviral CRISPR-Cas9 system (*Ran et al., 2013*), also failed to sustain the abscission checkpoint, as seen by a lack of an increase in midbody-connected cells when compared to non-targeting (NT) siRNA transfection, or control cell lines lacking guide RNAs against ULK3 (*Figure 2E*, lanes 8 and 10 vs lanes 2, 4, 6).

Another trigger for the abscission checkpoint is the presence of lagging chromosomes within the midbody (*Norden et al., 2006*; *Mendoza et al., 2009*; *Steigemann et al., 2009*), and we tested whether ULK3 was also required for this process. The presence of chromatin bridges within the midbody was monitored by time-lapse microscopy in cells expressing YFP-tagged lamina-associated polypeptide 2β (LAP2β) (*Steigemann et al., 2009*). As described previously for CHMP4C-depleted cells (*Carlton et al., 2012*), LAP2β bridges were also resolved more rapidly upon depletion of ULK3 (*Figure 2F* and *Figure 2—figure supplement 2A*, *Videos 4, 5*). In control cells, persisting chromosome bridges were resolved in 680 min, whereas in ULK3-depleted cells, this resolution time was dramatically reduced to 300 min. Thus, our experiments demonstrate that ULK3 plays an essential role in supporting the Aurora B-dependent abscission checkpoint in response to either defective nuclear pore assembly or lagging chromosomes. Besides delaying abscission, Aurora B activation also prevents cleavage furrow regression in cells that contain chromosome bridges (*Steigemann et al., 2009*). As expected, the frequency of furrow regression increased in cells with lagging chromosomes, but ULK3 depletion did not change this proportion (*Figure 2—figure supplement 2B,C*, *Videos 6, 7*). This result indicates that ULK3 does not control furrow regression, but instead likely functions downstream of Aurora B in delaying abscission in response to chromatin bridges.

## ULK3 functions in tension-dependent abscission regulation

A recent study revealed that the high-membrane tension of cells grown at low density delays abscission as compared to cells grown at high density (*Lafaurie-Janvore et al., 2013*). We, therefore, asked whether ULK3 also plays a role in this largely unexplored regulatory mechanism. Abscission events from *Figure 2A* were stringently segregated into two extreme groups: those for isolated cells (low density) and those for closely packed cells (high density). As reported previously, low-density cells showed significant abscission delays as compared to high-density cells (*Figure 2G*). In contrast, ULK3-depleted cells showed similar rapid abscission times regardless of the cell density (*Figure 2G*, lanes 3 and 4, *Videos 8–11*). Thus, ULK3 is also required to delay abscission in response to midbody tension.

## ESCRT-III binding and catalytic activity are required for ULK3 checkpoint function

We next performed mutational analyses to test the requirements for ULK3 to function in the abscission checkpoint. The faster abscission in ULK3-depleted cells was restored to control levels by stable expression of an siRNA-resistant wild-type ULK3 construct (ULK3R WT, *Figure 3A*, lane 3), but not by

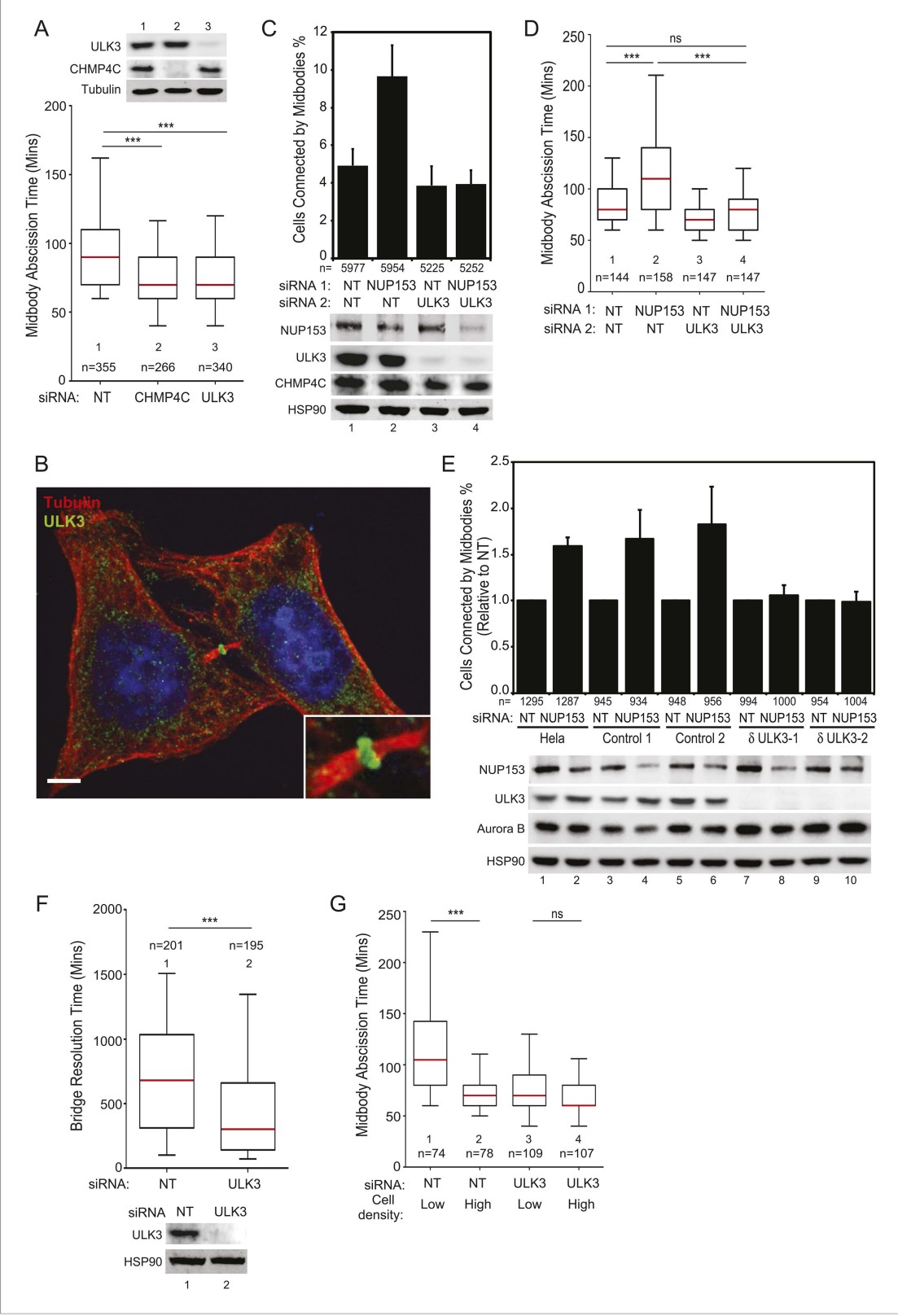

**Figure 2**. ULK3 regulates abscission timing. (**A**) Asynchronous cultures of HeLa mCherry-Tubulin cells were transfected with the specified siRNA. Midbody resolution times were calculated from three separate experiments (mean times ±SD were non-targeting (NT): 96 ± 36 min; CHMP4C: 74 ± 26 min; ULK3: 74 ± 31 min). (**B**) HeLa cells

*Figure 2. continued on next page*

*Figure 2. Continued*

were stained with Hoechst, α-Tubulin, and α-ULK3 antibody (Santa Cruz). ULK3 was observed at the Flemming body in 79% of observed midbodies (total n = 34). Image shows a representative example of Flemming body localization, with the ULK3 signal in green, Tubulin in red, and nuclei in blue. Bar = 5 μm. Inset shows an expanded view of the midbody. (**C** and **D**) HeLa mCherry-Tubulin cells transfected with NT or ULK3 siRNA were transfected with Nucleoporin 153 (NUP153) siRNA to trigger the abscission checkpoint. In (**C**), cells were fixed and stained with α-Tubulin antibody to visualize midbody-arrested cells. Data are represented as a mean percentage of midbodies ± SD from six separate experiments. NUP153 depletion levels as mean percentages from three independent experiments were lane 1: 100%, 2: 67%, 3: 108%, 4: 70%. In (**D**), midbody abscission times were analyzed (mean times are 1: 86 ± 22 min; 2: 118 ± 50 min; 3: 70 ± 15 min; 4: 81 ± 25 min). (**E**) Clonal HeLa cells stably expressing a lentiviral vector containing CRISPR-Cas9 with a guide RNA sequence targeting the ULK3 locus (δULK3-1 or δULK3-2) or without guide RNA (Controls) were transfected with NT or NUP153 siRNA as above and stained with α-Tubulin antibody to visualize midbody-arrested cells. Data are represented as the mean percentage increase in midbody-connected cells compared to the percentage of midbodies in NT-treated cells for each cell line ±SD from four separate experiments. Mean percentages of NUP153 depletion levels compared to HeLa + NT siRNA (lane 1) from four independent experiments were lane 1: 100%, 2: 52%, 3: 85%, 4: 52%, 5: 94%, 6: 51%, 7: 75%, 8: 48%, 9: 96%, 10: 48%. (**F**) HeLa YFP-lamina-associated polypeptide 2β (LAP2β) expressing cells were transfected with NT and ULK3 siRNA, and resolution times of intercellular chromatin bridges were quantified in three separate experiments (mean times are NT: 708 min; ULK3: 459 min). (**G**) Analysis of tension-dependent modulation of abscission time, mean times: 1: 120 ± 53 min; 2: 73 ± 19 min; 3: 77 ± 41 min; 4: 69 ± 20 min). Data in (**A**, **D**, **F**–**G**) are represented as box plots showing median abscission times (**A**, **D**, and **G**) or LAP2β bridge resolution times (**F**). Here and throughout, whiskers mark 5–95 percentiles, box edges represent the first and third quartiles, and red bars denote the median. n = total number of events counted per sample. Cell lysates in (**A**, **C**, **E**, and **F**) were examined by Western blot using indicated antibodies. See also *Figure 2—figure supplements 1* and *2*, and *Videos 1–11*.

The following figure supplements are available for figure 2:

**Figure supplement 1**. ULK3 is not required for endosomal sorting or HIV-1 budding, but does regulate abscission timing.

**Figure supplement 2**. LAP2β-positive intercellular chromatin bridges resolve faster in ULK3-depleted cells.

expression of ULK3 proteins with inactivating mutations in the MIM1-binding sites of either (or both) MIT domains (ULK3$^R$ V338D, ULK3$^R$ M434D and ULK3$^R$ V338D, M434D; *Figure 3A*, lanes 4–6) or with an inactivating mutation in the ATP-binding site (K44H) of the kinase domain (*Chan et al., 2009*) (*Figure 3E*, lane 4 vs 3). Similarly, stable overexpression of these MIT (*Figure 3B*, *Videos 12, 13*) or catalytic (*Figure 3F*, lane 3 vs 2, *Video 14*) mutants did not change the timing of abscission, whereas ULK3$^R$ WT overexpression induced abscission delays (*Figure 3B*, *Videos 15–17*). The ULK3 MIT mutations did not alter ULK3 kinase activity (*Figure 1—figure supplement 1G*), and we, therefore, attribute their effects in abscission specifically to reductions in ESCRT-III binding. The efficacy of the K44H mutation was confirmed by the lack of auto-phosphorylation activity of GST-ULK3 K44H in an in vitro kinase assay (*Figure 3C*), and retention of ESCRT-III binding activity was confirmed by Y2H (*Figure 3D*). Thus, ULK3 modulates midbody resolution timing through both its kinase activity and ESCRT-III interactions.

## Effects of ULK3 on the abscission machinery

The functional links with ESCRT-III subunits suggested that catalytically active ULK3 might regulate abscission timing by influencing ESCRT-III polymerization during cytokinesis. We tested this hypothesis by analyzing the midbody location of endogenous IST1, which is essential for abscission (*Agromayor et al., 2009*; *Bajorek et al., 2009a*). ULK3 overexpression altered IST1 midbody localization significantly, as indicated by a 1.5-fold increase in the proportion of IST1 found at the midbody rings and a marked ninefold increase in the proportion of midbodies containing a single ring of IST1 at the Flemming body (*Figure 4A*). Importantly, HA-ULK3 was seen to co-localize with IST1 at this Flemming body ring (in 42% of observed midbodies) (*Figure 4B*). Additionally, ULK3 and IST1 co-localized at midbody rings on both sides of the Flemming body (58% of cases) (*Figure 4B*), which could be a transient ULK3 localization rarely observed in steady-state conditions exacerbated in the overexpression context (compare to endogenous staining of ULK3 in *Figure 2B*).

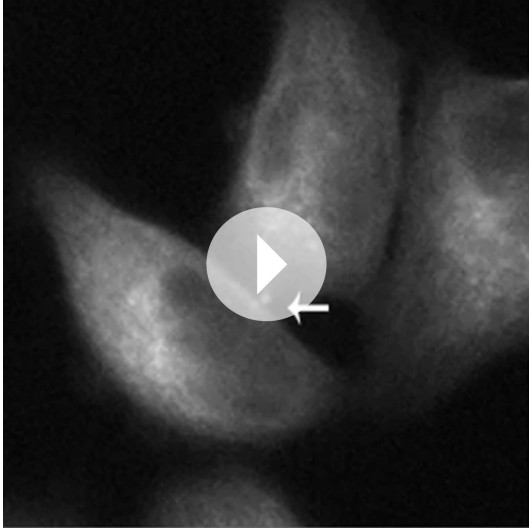

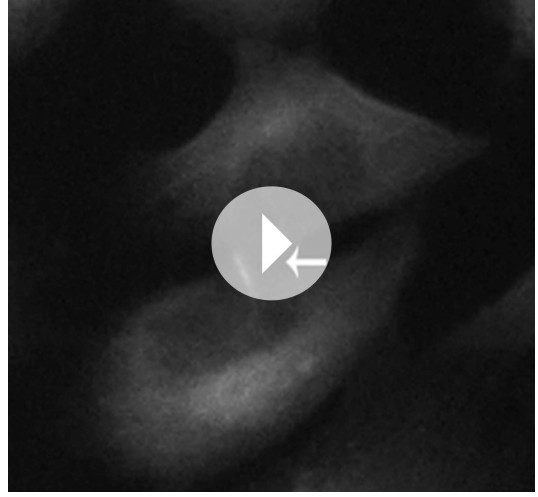

**Video 1.** Representative example of asynchronous HeLa mCherry-Tubulin cells treated with non-targeting (NT) siRNA. Midbody resolution is indicated with an arrow. Abscission time is 90 min. Related to *Figure 2* and *Figure 2—figure supplement 1*.

**Video 2.** Representative example of asynchronous HeLa mCherry-Tubulin cells treated with ULK3 siRNA. Midbody resolution is indicated with an arrow. Abscission time is 60 min. Related to *Figure 2* and *Figure 2—figure supplement 1*.

As a second marker for ESCRT-III polymers, we investigated whether ULK3 overexpression altered the midbody localization of GFP-CHMP4B expressed at endogenous levels (*Jouvenet et al., 2011*). As with IST1, the proportion of midbodies exhibiting single Flemming body rings of GFP-CHMP4B also increased 20-fold upon ULK3 overexpression (*Figure 4—figure supplement 1A,B*). In contrast, overexpression of either ESCRT-binding deficient (M434D) or kinase-defective (K44H) ULK3 had no effect on GFP-CHMP4B localization (*Figure 4—figure supplement 1A*). Thus, the ability of ULK3 constructs to induce aberrant ESCRT-III

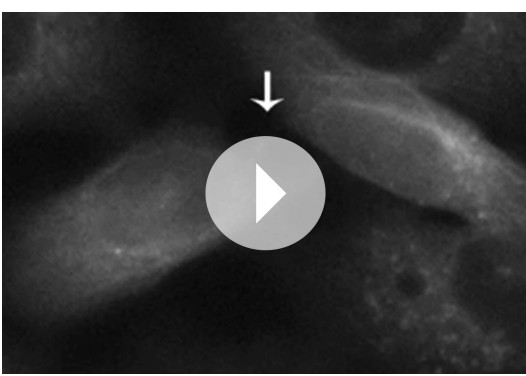

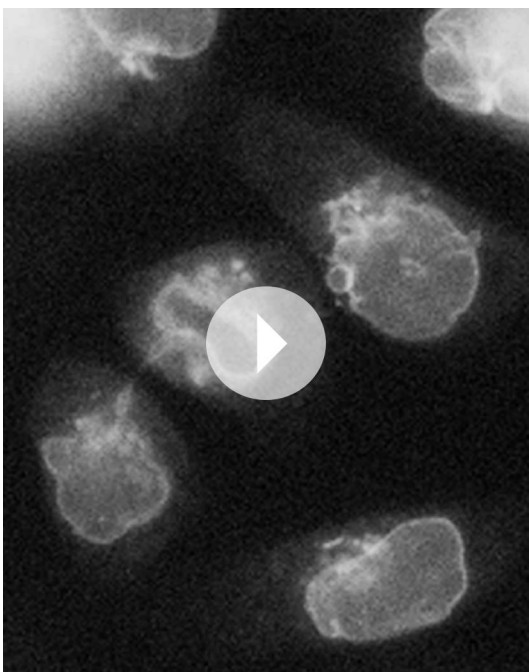

**Video 3.** Representative example of asynchronous HeLa mCherry-Tubulin cells treated with ULK3 siRNA. Midbody resolution is indicated with an arrow. Abscission time is 60 min. Related to *Figure 2* and *Figure 2—figure supplement 1*.

**Video 4.** Representative example of asynchronous HeLa cells stably expressing YFP-LAP2β transfected with NT siRNA. Chromatin bridge resolution time is 510 min. Bridge resolution is indicated with an arrow. Related to *Figure 2* and *Figure 2—figure supplement 2*.

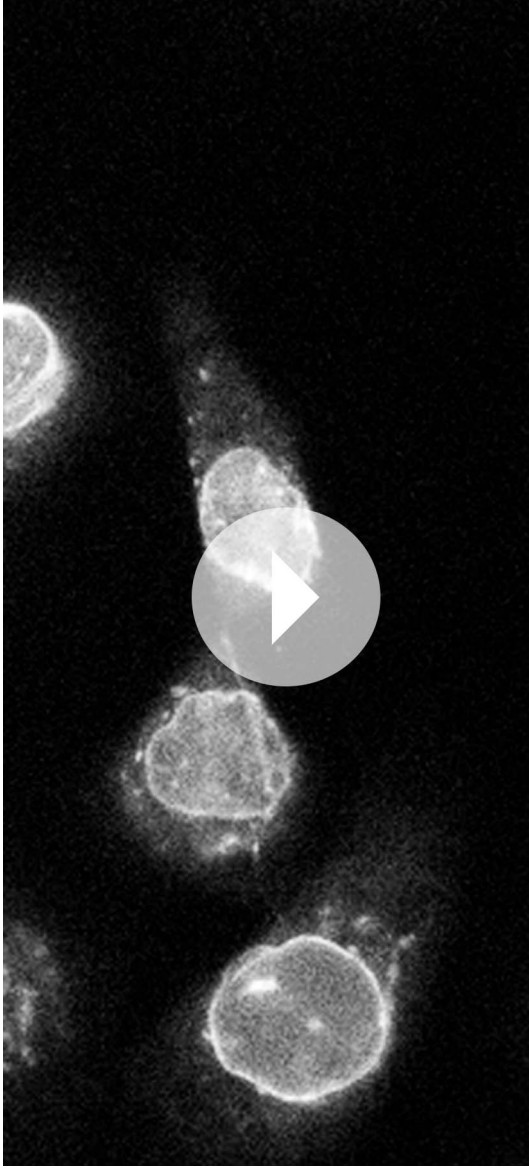

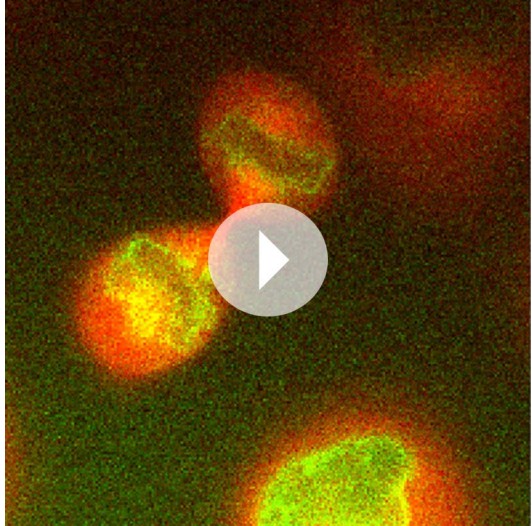

**Video 6.** Representative example of asynchronous HeLa cells stably expressing YFP-LAP2β and mCherry-Tubulin, containing an intercellular chromatin bridge that resolves in 850 min when the daughter cells faithfully divide. Bridge resolution is indicated with an arrow. Related to *Figure 2—figure supplement 2*.

**Video 5.** Representative example of asynchronous HeLa cells stably expressing YFP-LAP2β transfected with ULK3 siRNA. Chromatin bridge resolution time is 110 min. Bridge resolution is indicated with an arrow. Related to *Figure 2* and *Figure 2—figure supplement 2*.

localization correlates with their ability to delay abscission. Both WT and K44H ULK3 co-localized with GFP-CHMP4B polymers at the intercellular bridge (*Figure 4—figure supplement 1C,D*), while only ULK3 WT co-localized with GFP-CHMP4B at the Flemming body (39% vs 61% of ULK3/GFP-CHMP4B at the midbody rings) (*Figure 4—figure supplement 1C*), more closely reflecting the Flemming body localization observed for endogenous ULK3 in a context without overexpression (compare to *Figure 2B*). This observation suggests that ULK3 does not delay abscission by directly blocking or outcompeting ESCRT-III interactions required for polymerization. Instead, ULK3 apparently induces abscission delays and ESCRT-III mislocalization by phosphorylating substrates important for abscission completion. Taken together, these data show that catalytically active ULK3 changes the steady-state distribution of ESCRT-III proteins at the midbody, suggesting that the abscission delays induced by ULK3 are due, at least in part, to defective ESCRT-III polymerization.

## ULK3 phosphorylates ESCRT-III subunits

We next tested whether ESCRT-III subunits are phosphorylated by ULK3. An in vitro kinase assay using recombinant ULK3 on immunoprecipitated HA-ESCRT-III proteins showed phosphorylation of CHMP1A, CHMP1B, CHMP2A, and IST1 (*Figure 5A*, middle blot, lanes 2, 3, 4, and 10), but not other ESCRT-III subunits. Analogous phosphorylation events were recapitulated in cells overexpressing OSF-ULK3 and Myc-tagged ESCRT-III proteins. The enhanced separation of

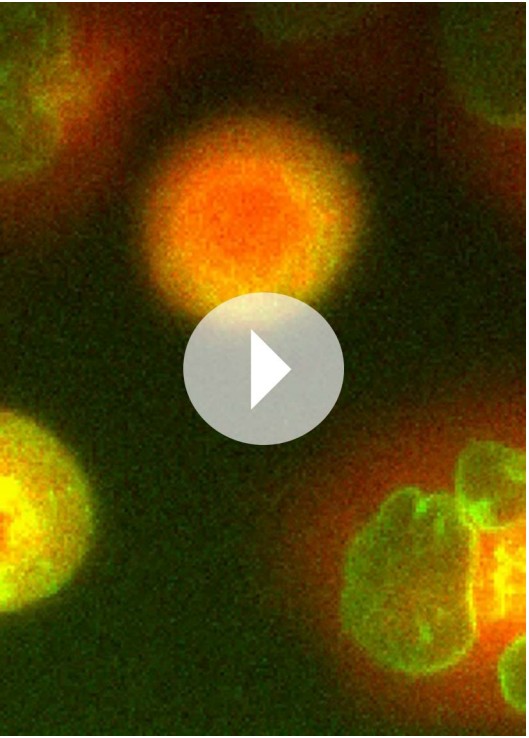

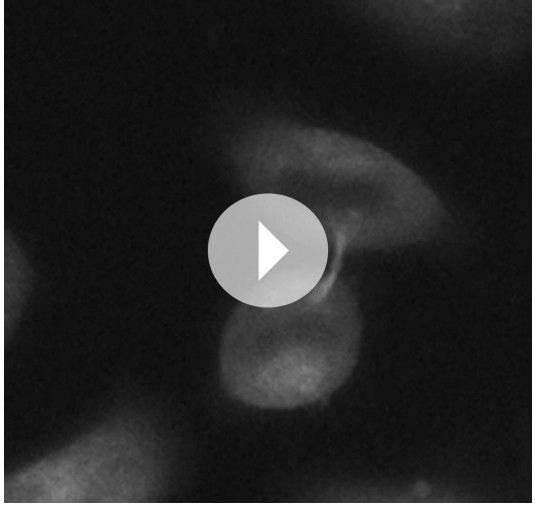

**Video 8.** Representative example of low-density asynchronous HeLa mCherry-Tubulin cells treated with NT siRNA. Midbody resolution is indicated with an arrow. Abscission time is 170 min. Related to *Figure 2*.

**Video 7.** Representative example of asynchronous HeLa cells stably expressing YFP-LAP2β and mCherry-Tubulin, containing an intercellular chromatin bridge that does not resolve and the cleavage furrow regresses. Related to *Figure 2—figure supplement 2*.

phosphorylated species by electrophoresis on Phos-tag SDS-PAGE gels revealed ULK3-dependent phosphorylation of IST1, CHMP1A, CHMP1B, and CHMP2A, but not CHMP3 (*Figure 5B*, lane 2 and *Figure 5—figure supplement 1A*, lanes 2, 8, 14, and 20). Phosphorylation was confirmed by sensitivity to calf intestinal phosphatase (CIP) treatment (*Figure 5B*, lanes 4–6 and *Figure 5—figure supplement 1A*, lanes 4–6, 10–12, 16–18, and 22–24) and by expressing the inactive ULK3 K44H protein as a control (*Figure 5B*, lane 3 and *Figure 5—figure supplement 1A*, lanes 3, 9, and 15). These data demonstrate that ULK3 ESCRT-III binding partners are also substrates for phosphorylation.

## Identification of ULK3 phosphorylation sites on IST1

ULK3 phosphorylation of IST1 was characterized by mass spectrometry. To maximize IST1 phosphorylation, we co-overexpressed Myc-IST1 and OSF-ULK3 in 293T cells, affinity purified the IST1/ULK3 complexes, and determined the masses of intact IST1 proteins by electrospray ionization (ESI) mass-spectrometry. We observed four different IST1 species whose masses differed by ∼80 Da, the mass change corresponding to phosphorylation (*Figure 5—figure supplement 1B*). The different masses corresponded to Myc-IST1 proteins with 1–4 phosphates (and N-terminal acetylation). These assignments were confirmed by measuring the intact mass of Myc-IST1 co-expressed with a catalytically inactive ULK3 protein (K139R) (*Maloverjan et al., 2010*) and CIP treated (*Figure 5—figure supplement 1B*).

Phosphorylation sites were mapped by ESI/MS/MS analyses of IST1 peptides from tryptic and chymotryptic digests. These analyses confirmed that Myc-IST1 is N-terminally acetylated and identified four phosphorylated serine residues: S4, S99, S153, and S214. These sites are upstream of the IST1 MIM elements, either within the core ESCRT-III four-helix bundle (*Muziol et al., 2006*; *Bajorek et al., 2009b*) (residues 1–189; S4, S99, and S153) or in a flexible region flanking the core (S214; *Figure 5C*).

We next compared the phosphorylation status of endogenous IST1 in cycling cells vs Nocodazole-arrested cells. Mitotic arrest was confirmed by the presence of Histone 3 phosphorylation (P-H3), a marker for mitotic chromosome condensation (*Hendzel et al., 1997*) (*Figure 5D*). Nocodazole

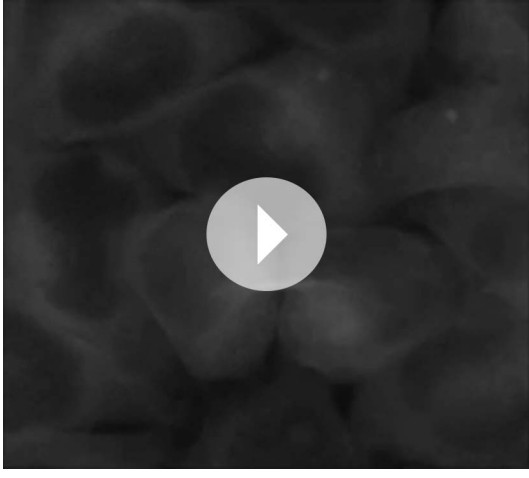

**Video 9.** Representative example of high-density asynchronous HeLa mCherry-Tubulin cells treated with NT siRNA. Midbody resolution is indicated with an arrow. Abscission time is 60 min. Related to *Figure 2*.

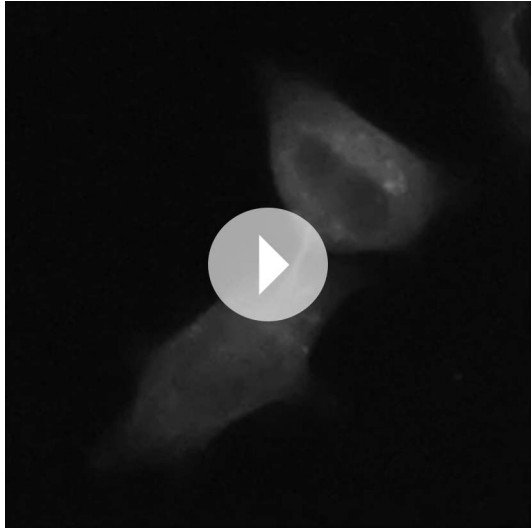

**Video 10.** Representative example of low-density asynchronous HeLa mCherry-Tubulin cells treated with ULK3 siRNA. Midbody resolution is indicated with an arrow. Abscission time is 60 min. Related to *Figure 2*.

treatment induced accumulation of at least two different phosphorylated IST1 species (*Figure 5D*). A similar banding pattern was observed when endogenous IST1 was depleted and replaced by an siRNA-resistant wild-type IST1 construct (IST1$^R$ WT, lane 2 vs 4). However, the lowest mobility IST1 species was eliminated when the four mapped phosphorylated serine residues were mutated to alanine (IST1$^R$ 4SA, *Figure 5D*, lane 6). The IST1$^R$ 4SA mutant still formed intermediate mobility species, which suggests that additional site(s) on IST1 may be phosphorylated by ULK3 or other mitotic kinases. Formation of the lowest mobility phosphorylated IST1 species was ULK3-dependent, as it was also eliminated by depletion of ULK3 from Nocodazole-treated cells (*Figure 5E*, lane 5 vs 4). Taken together, these results indicate that ULK3 phosphorylates endogenous IST1 during mitosis.

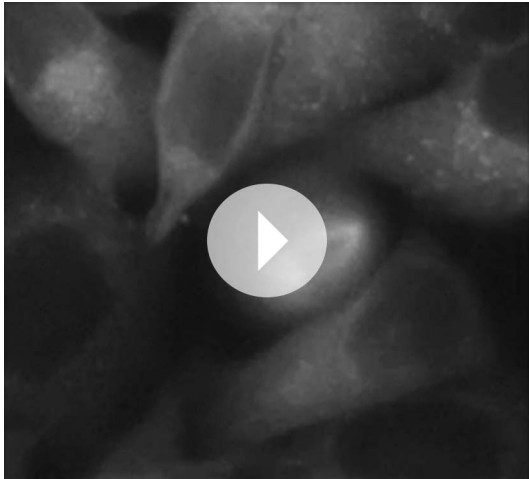

**Video 11.** Representative example of high-density asynchronous HeLa mCherry-Tubulin cells treated with ULK3 siRNA. Midbody resolution is indicated with an arrow. Abscission time is 60 min. Related to *Figure 2*.

## ULK3-dependent phosphorylation of IST1 is required for abscission checkpoint signaling

To evaluate the functional consequences of IST1 phosphorylation by ULK3, we tested the ability of the IST1$^R$ 4SA mutant to support the abscission checkpoint. These experiments revealed that NUP153 depletion failed to trigger the abscission checkpoint when endogenous IST1 was replaced with IST1$^R$ 4SA (*Figure 6A*, lane 6 vs 2), whereas the checkpoint was fully functional when IST1$^R$ WT was used in this assay (*Figure 6A*, lane 4 vs 2). Similarly, intercellular chromatin bridges in cells depleted of endogenous IST1 but expressing IST1$^R$ 4SA resolved significantly faster than chromatin bridges in control cells or cells that expressed IST1$^R$ WT (*Figure 6B*). Importantly, both IST1$^R$ WT and IST1$^R$ 4SA were able to restore the abscission defects induced by IST1 depletion (*Figure 6C*, lanes 3–4 vs lane 2, and *Figure 6D*), indicating that IST1$^R$ 4SA is fully

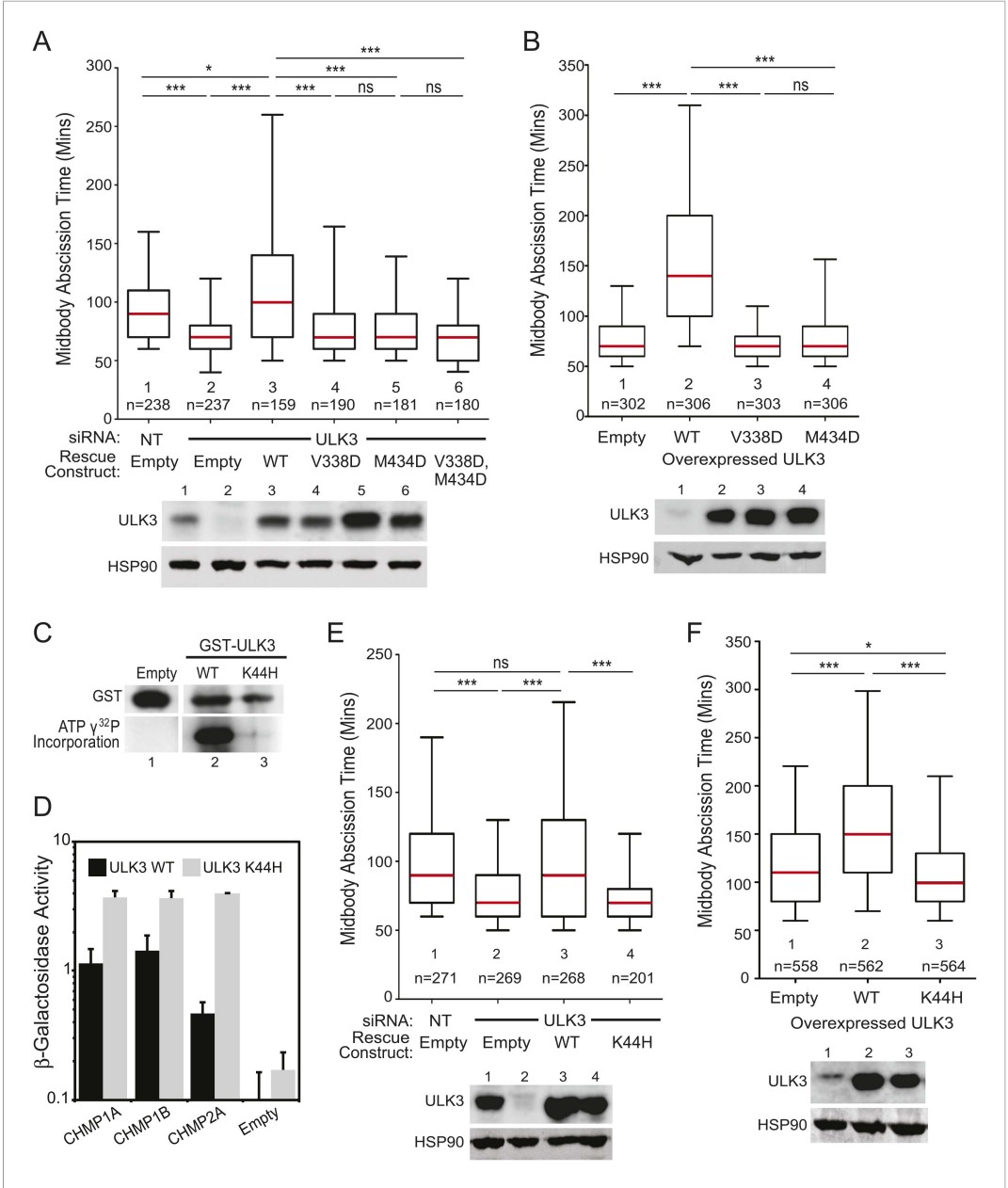

**Figure 3**. ESCRT-III binding and kinase activity are required for ULK3 function. (**A**) HeLa mCherry-Tubulin cells stably expressing empty vector or siRNA-resistant ULK3 (ULK3[R]) were transfected with the indicated siRNA. Midbody resolution times were calculated in three separate experiments (mean times were 1: 95 ± 32 min; 2: 71 ± 25 min; 3: 115 ± 61 min; 4: 80 ± 38 min; 5: 77 ± 34 min; 6: 74 ± 31 min). (**B**) HeLa mCherry-Tubulin cells expressing empty vector, ULK3 WT, V338D, or M434D were imaged live. Midbody resolution times were analyzed from three separate experiments (mean times were 1: 81 ± 29 min; 2: 159 ± 88 min; 3: 73 ± 23 min; 4: 83 ± 37 min). (**C**) In vitro kinase assay showing the auto-phosphorylation activity of GST-ULK3 WT and GST-ULK3 K44H. (**D**) Yeast two-hybrid (Y2H) assay with ULK3 WT and ULK3 K44H fused to VP16 binding to ESCRT-III proteins fused to Gal4. Data are represented as mean β-galactosidase activity ±SD from triplicate measurements of two separate experiments. (**E**) HeLa mCherry-Tubulin cells expressing either empty vector or ULK3[R] constructs were transfected with the indicated siRNA. Midbody resolution times were analyzed in three independent experiments (mean times were 1: 100 ± 42 min; 2: 77 ± 34 min; 3: 104 ± 57 min; 4: 74 ± 31 min). (**F**) HeLa mCherry-Tubulin cells expressing empty vector, ULK3 WT, or K44H were imaged live. Midbody resolution times were calculated in four separate experiments (mean times are 1: 122 ± 60 min; 2: 161 ± 73 min; 3: 113 ± 47 min). Cell lysates in (**A**, **B**, **E**, and **F**) were examined by Western blot using the indicated antibodies. See also *Videos 12–17*.

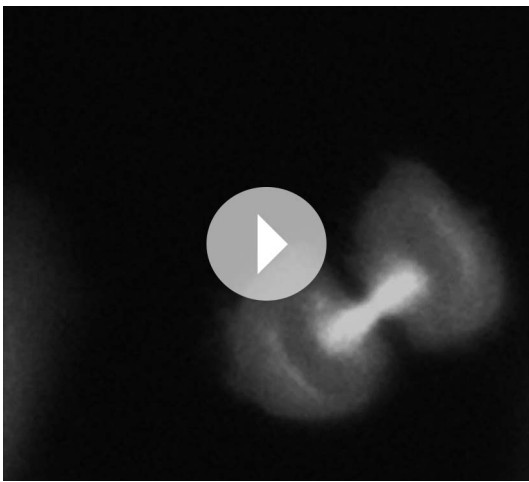

**Video 12.** Representative example of asynchronous HeLa cells stably expressing mCherry-Tubulin and ULK3 V338D. Midbody resolution is indicated with an arrow. Abscission time is 80 min. Related to *Figure 3*.

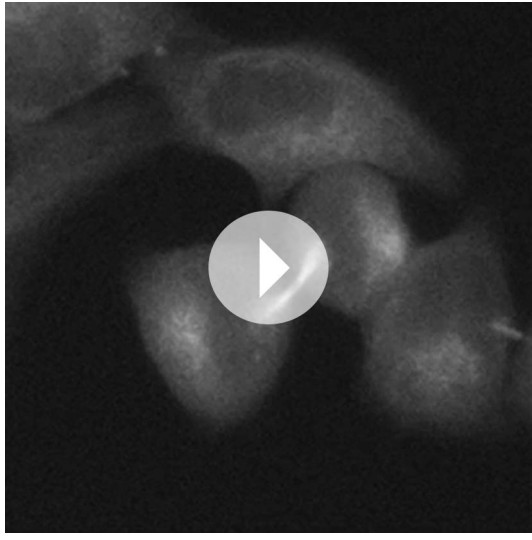

**Video 13.** Representative example of asynchronous HeLa cells stably expressing mCherry-Tubulin and ULK3 M434D. Midbody resolution is indicated with an arrow. Abscission time is 70 min. Related to *Figure 3*.

functional for cytokinesis under steady-state conditions. Similarly, IST1$^R$ 4SA retained its interactions with all known IST1-binding partners in Y2H assays (*Figure 6—figure supplement 1A*). It was notable that the IST1$^R$ 4SA could also still support abscission delays when the checkpoint was induced by low-cell tension. Specifically, the effects on tension-mediated abscission regulation were similar when cells expressed IST1$^R$ WT or IST1$^R$ 4SA (*Figure 6—figure supplement 1B*). Thus, the IST1$^R$ 4SA protein did not support the abscission checkpoint in response to chromatin within the midbody and nuclear pore disruption but retained other IST1 binding and abscission functions, including regulation of abscission timing in response to midbody tension. In contrast, a phosphomimetic IST1 mutant (IST1$^R$ 4SE: S4E, S99E, S153E, S214E) failed to restore abscission in IST1-depleted cells, as reflected by the accumulation of multinucleated and

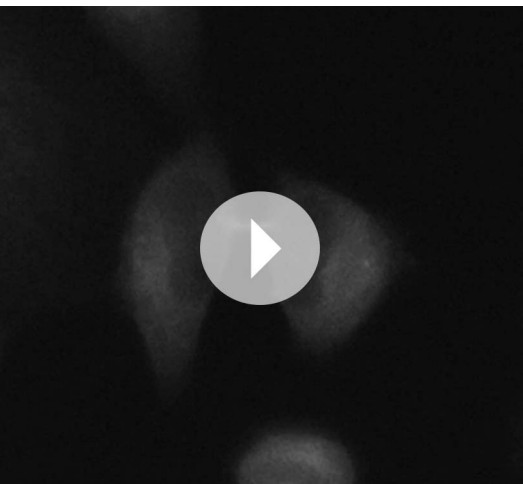

**Video 14.** Representative example of asynchronous HeLa cells stably expressing mCherry-Tubulin and ULK3 K44H. Midbody resolution is indicated with an arrow. Abscission time is 80 min. Related to *Figure 3*.

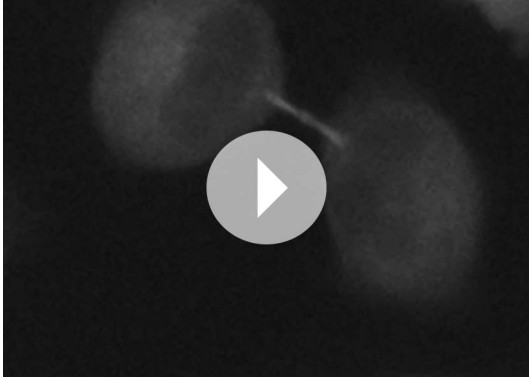

**Video 15.** Representative example of asynchronous HeLa cells stably expressing mCherry-Tubulin and empty vector. Midbody resolution is indicated with an arrow. Abscission time is 100 min. Related to *Figure 3*.

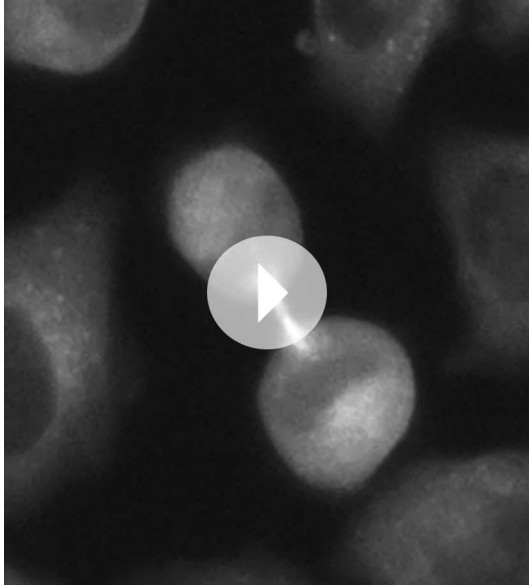

**Video 16.** Representative example of asynchronous HeLa cells stably expressing mCherry-Tubulin and ULK3 WT. Midbody resolution is indicated with an arrow. Abscission time is 160 min. Related to *Figure 3*.

**Video 17.** Representative example of asynchronous HeLa cells stably expressing mCherry-Tubulin and ULK3 WT. Midbody resolution is indicated with an arrow. Abscission time is 290 min. Related to *Figure 3*.

midbody-arrested cells (*Figure 6C*, lane 5 and *Figure 6D*, bottom panel). Furthermore, live-cell imaging confirmed that even when cells expressing IST1$^R$ 4SE did complete abscission, they did so more slowly than cells expressing IST1$^R$ WT or 4SA (*Figure 6E*, *Videos 18–22*). Collectively, these experiments indicate that ULK3 phosphorylation of IST1 delays abscission as an essential component of the abscission checkpoint in the presence of chromatin bridges within the midbody or incomplete nuclear pore assembly.

We next used a co-precipitation approach to test whether phosphorylation altered the interactions of IST1 with its binding partners (*Figure 6F*). These experiments revealed that the phosphomimetic mutations enhanced binding of YFP-IST1 4SE to both GST-VPS4 and GST-LIP5 as compared to YFP-IST1 WT and YFP-IST1 4SA (*Figure 6F*, lanes 20 and 24). In contrast, the interaction of YFP-IST1 4SE with GST-CHMP1B was unchanged (*Figure 6F*, lane 12), and the interaction with GST-CHMP2A remained negative. Thus, the enhanced binding effects of the IST1 phosphomimetic mutations were specific for VPS4 and its co-activator LIP5. These results suggest that ULK3 phosphorylation of IST1 may delay abscission by regulating the interaction of IST1 with late-acting components of the ESCRT machinery.

## ULK3 and CHMP4C are functionally interconnected within the Aurora B-dependent abscission control pathway

The phenotypic similarities between CHMP4C and ULK3 in regulating the abscission checkpoint suggested connections between these proteins. Consistent with this idea, CHMP4C depletion eliminated the IST1 hyperphosphorylation induced by Nocodazole treatment, mimicking the effects of ULK3 depletion (*Figure 5E*, lane 6). This observation reveals a requirement for CHMP4C in IST1 phosphorylation and suggests a functional link between ULK3 and CHMP4C. To examine whether ULK3 and CHMP4C function in the same regulatory pathway, we tested whether ULK3 is required for the abscission delays induced by overexpression of GFP-CHMP4C (*Carlton et al., 2012*). GFP-CHMP4C overexpression increased the steady-state number of cells connected by midbodies, but ULK3 depletion reverted this phenotype to control levels (*Figure 7A*, lane 5 vs 4). Comparable effects were observed for cells treated with CHMP4C siRNA, which depleted both endogenous and exogenous CHMP4C (lane 6). Interestingly, ULK3 depletion did not alter GFP-CHMP4C localization within the central region of the midbody (*Figure 7B*), suggesting that ULK3 is not required for

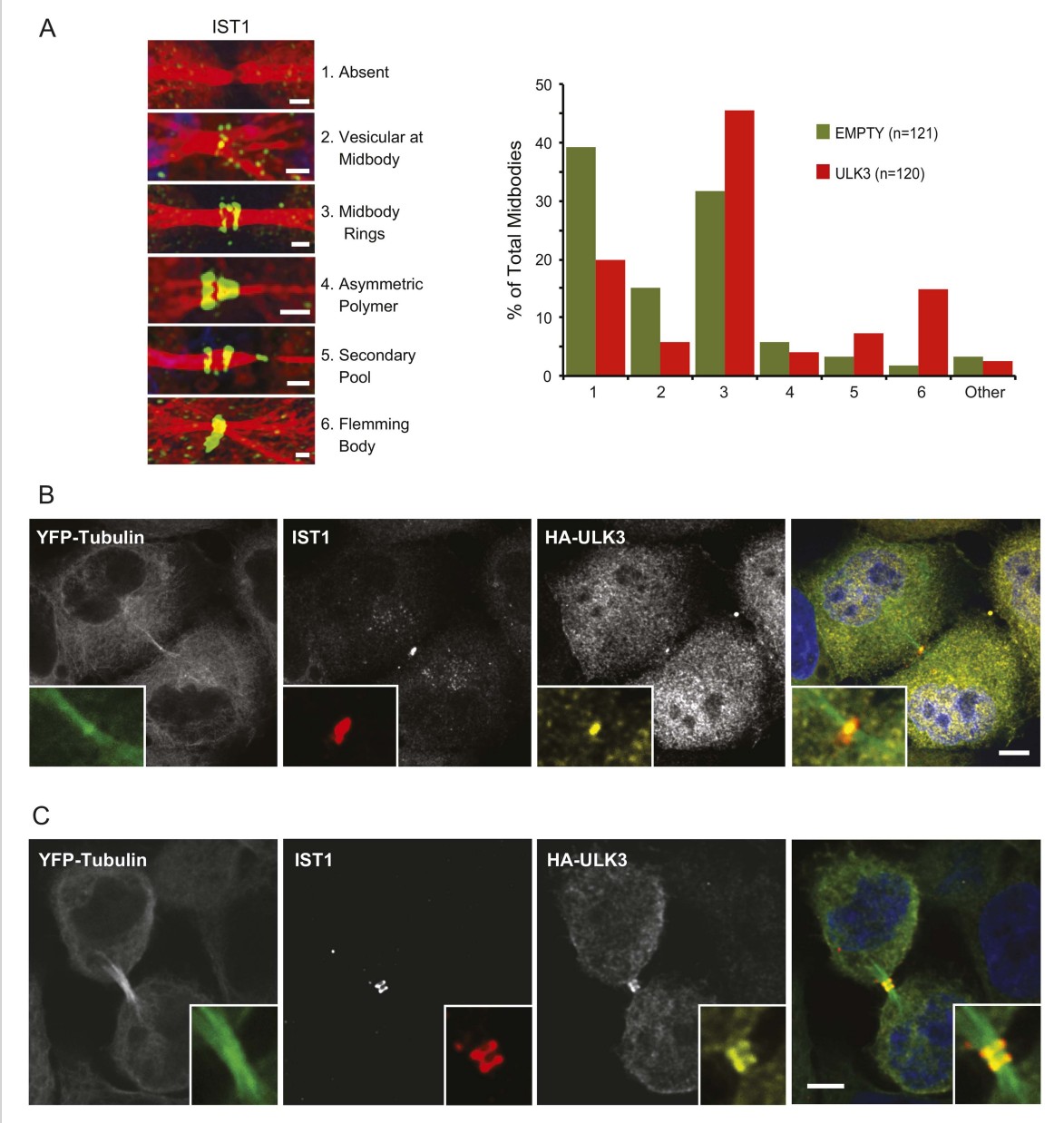

**Figure 4**. Effects of ULK3 on IST1 localization at the midbody. (**A**) HeLa mCherry-Tubulin cells stably expressing empty vector or ULK3 WT were stained with Hoechst and α-IST1 antibody. Cells connected by midbodies were classified by the localization pattern of IST1 as shown on the left panel, where Tubulin is red and IST1 is green. Bars = 1 μm. Data are represented as percentage of midbodies observed for each category from three separate experiments; n = total number of scored midbodies. Cells connected by chromatin bridges or multinucleated were excluded. (**B** and **C**) Panels show a representative example of Flemming body localization (**B**) or midbody rings (**C**), where the IST1 signal (middle left) co-localizes with HA-ULK3 (middle right) in HeLa YFP-Tubulin (left) cells expressing HA-ULK3 stained with α-IST1 and α-HA antibodies. ULK3 and IST1 co-localization was observed in 100% of midbodies with Flemming body localization for IST1 (n = 48). Merged channels are shown on the right (green: Tubulin, red: IST1, yellow: HA, blue: nuclei). Magnifications of the midbody are shown for each channel. Bar = 5 μm. See also **Figure 4—figure supplement 1**.

The following figure supplement is available for figure 4:

**Figure supplement 1**. Effects of ULK3 on the abscission machinery.

transmission of the Aurora B signal that targets CHMP4C to the Flemming body (*Carlton et al., 2012*). Endogenous ULK3 did co-localize with GFP-CHMP4C at the Flemming body, however, further supporting a functional connection between these proteins (*Figure 7C*).

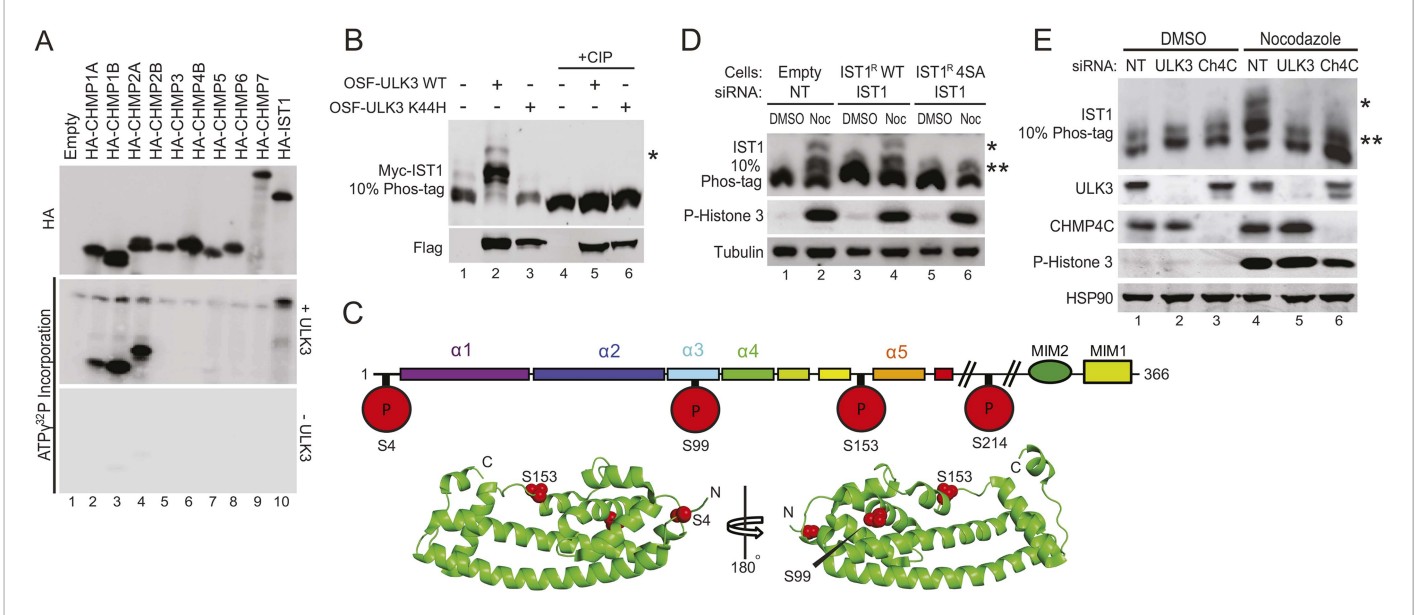

**Figure 5**. Identification of ULK3 phosphorylation sites within IST1. (**A**) In vitro kinase assay with recombinant ULK3 protein (middle) on immunoprecipitated HA-ESCRT-III proteins expressed in 293T cells. Western blot with α-HA antibody shows immunoprecipitated ESCRT-III proteins (top). (**B**) 293T cells co-transfected with Myc-IST1 and either empty vector, OSF-ULK3 WT or OSF-ULK3 K44H. Lysates were electrophoresed on a 10% Phos-tag gel to separate phosphorylated species (denoted by 'asterisk'). Lysates were treated with calf intestinal phosphatase (CIP, lanes 4–6). (**C**) Identified phosphorylation sites mapped onto the IST1 domain structure (above) and onto the crystal structure of the IST1 N-terminal domain (below, PDB 3FRR) 39. (**D**) HeLa cells stably expressing empty vector, siRNA-resistant IST1 (IST1$^R$) WT, or IST1$^R$ 4SA were transfected with siRNA and treated overnight with DMSO or Nocodazole. (**E**) HeLa cells were transfected with siRNA prior to overnight treatment with DMSO or Nocodazole. In (**D** and **E**), cell lysates were resolved on 10% Phos-tag gels and analyzed by western blot, where 'asterisks' denote bands corresponding to IST1 phosphorylated species, sensitive (*) or insensitive (**) to the 4SA mutation or ULK3/CHMP4C depletion. See also *Figure 5—figure supplement 1*.

The following figure supplement is available for figure 5:

**Figure supplement 1**. ULK3 phosphorylation of ESCRT-III proteins.

The effects of ULK3 on GFP-CHMP4C-dependent abscission delays were confirmed by live imaging of cells co-expressing mCherry-Tubulin. In agreement with the fixed-cell data, ULK3 depletion reverted CHMP4C-induced delays in midbody resolution to control levels (defined by cells expressing GFP-CHMP4B, *Figure 7D*, *Videos 23–25*). In a reciprocal experiment, CHMP4C depletion attenuated abscission delays imposed by ULK3 overexpression, although in this case the effect was incomplete (*Figure 7E*, *Videos 26–28*). ULK3-induced abscission delays were abolished upon treatment of cells at midbody stage with an Aurora B inhibitor (ZM447439) (*Ditchfield et al., 2003*). ZM447439-treated cells rapidly underwent abscission, implying that Aurora B activity is required in order for ULK3 to delay abscission (*Figure 7F*). These data indicate that ULK3 and CHMP4C act together in regulating abscission as part of the Aurora B-dependent abscission control pathway. The requirement for CHMP4C in abscission delay can be partially alleviated in cells overexpressing ULK3, however, suggesting that excess ULK3 can overcome upstream CHMP4C functions in regulating midbody resolution.

We next assessed the effect of ULK3-mediated phosphorylation of IST1 and CHMP4C in the context of the Aurora B-dependent abscission pathway. As shown in *Figure 7G*, the lowest mobility IST1 species observed in mitotically arrested cells was eliminated upon treatment with Aurora B inhibitor (lane 4 vs 3), whereas the intermediate mobility species was not altered by this treatment. These effects are similar to those observed for the IST1$^R$ 4SA mutant or upon ULK3 depletion. These observations suggest that the low-mobility species is required for abscission checkpoint function and that additional site(s) on IST1 could be phosphorylated by ULK3 or other mitotic kinases in an Aurora B-independent manner. We, therefore, tested whether immunoprecipitated HA-tagged CHMP4

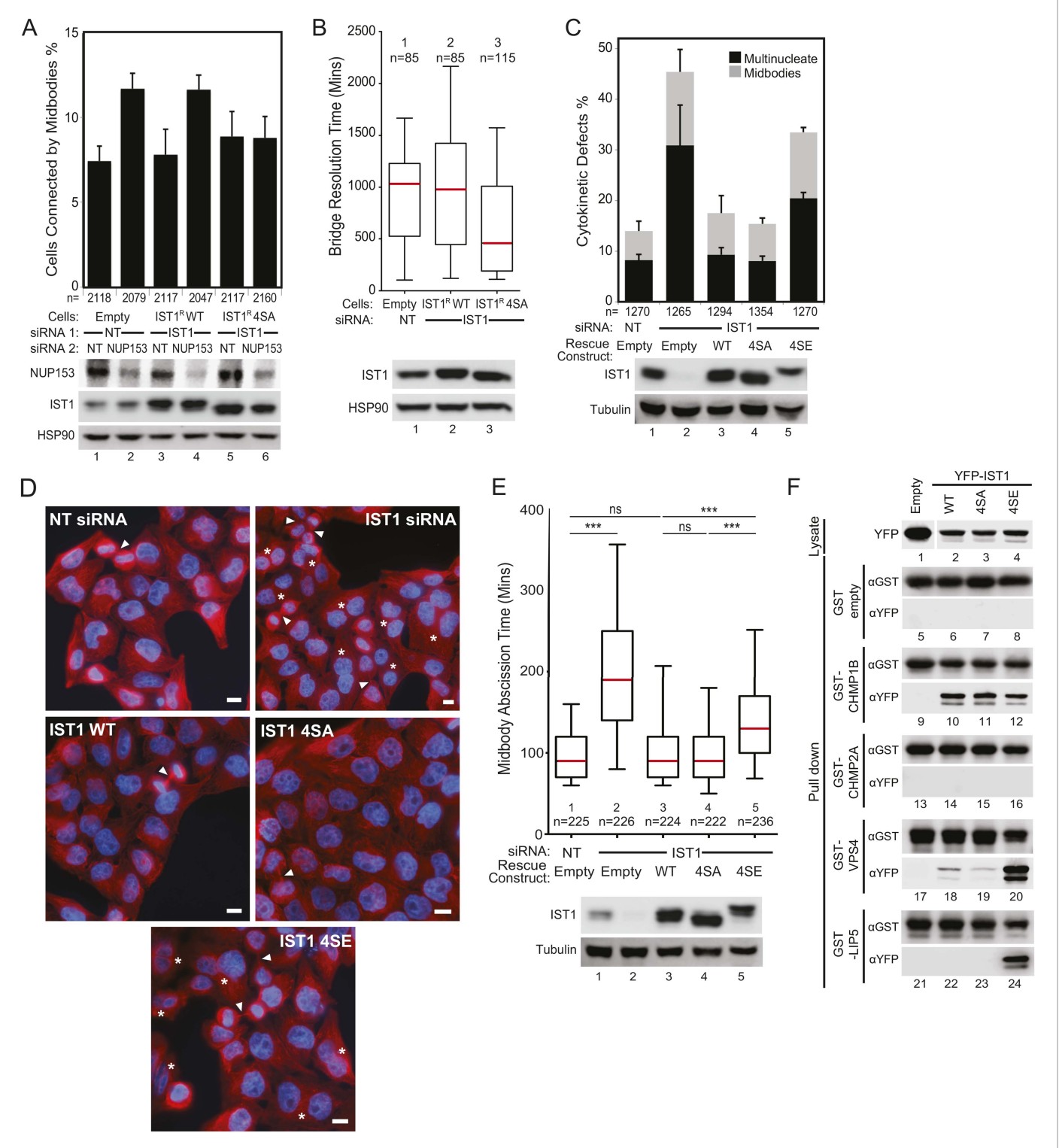

**Figure 6**. ULK3 phosphorylation of IST1 is required to sustain the abscission checkpoint and inhibits IST1 function in abscission. (**A**) HeLa cells stably expressing empty vector, IST1ᴿ WT, or IST1ᴿ 4SA were co-transfected with the indicated siRNA and NUP153 siRNA to trigger the abscission checkpoint. Cells were fixed and stained with Hoechst and α-Tubulin antibody to visualize nuclei and midbodies, respectively. Data are represented as mean percentage of midbodies ±SD from four separate experiments, n = total number of scored midbodies. NUP153 depletion levels as a mean percentage quantified by Western blot from two independent experiments were lane 1: 100%, 2: 40%, 3: 85%, 4: 39%, 5: 120%, 6: 46%. (**B**) HeLa YFP-LAP2β cells expressing empty vector, IST1ᴿ WT, or IST1ᴿ 4SA were transfected with NT or IST1 siRNA, and resolution times of intercellular chromatin bridges were
*Figure 6. continued on next page*

*Figure 6. Continued*

quantified in two separate experiments (mean times are 1: 906 min; 2: 965 min: 3: 638 min). (**C** and **D**) HeLa cells stably expressing empty vector, IST1$^R$ WT, IST1$^R$ 4SA, or IST1$^R$ 4SE were transfected with the indicated siRNA, fixed and stained with α-Tubulin antibody to visualize cytokinetic defects. Data in (**C**) are represented as mean percentage of midbody-arrested or multinucleated cells ±SD from three separate experiments. Microscopy images in (**D**) are representative examples from each sample in (**C**), shown as merged images of Tubulin (red) and nuclei (blue). 'Asterisks' and 'arrowheads' denote multinucleated and midbody-connected cells, respectively. Bars = 10 μm. (**E**) HeLa YFP-Tubulin cells stably expressing empty vector, IST1$^R$ WT, IST1$^R$ 4SA, or IST1$^R$ 4SE were transfected with the indicated siRNA. Live-cell imaging revealed midbody resolution times from two separate experiments (mean times were 1: 97 ± 33 min; 2: 204 ± 95 min; 3: 103 min ±48; 4: 98 ± 38 min; 5: 145 ± 65 min). Cell lysates in (**A**, **B**, **C**, and **D**) were examined by Western blot using the indicated antibodies. (**F**) Co-precipitation assay from 293T cells co-transfected with the indicated YFP and GST fusions. 10% of the volume eluted from the beads was analyzed by Western blot with α-YFP antibody to visualize bound proteins and α-GST as a control for pull-down efficiency. α-YFP Western blot on input lysates (top) is representative for all experiments, and empty YFP band was cropped from the same blot. See also *Figure 6—figure supplement 1* and *Videos 18–22*.

The following figure supplement is available for figure 6:

**Figure supplement 1**. The IST1 phosphorylation 4SA mutant retains IST1 binding and abscission functions.

proteins could be phosphorylated by ULK3 in vitro. CHMP4C was phosphorylated by recombinant ULK3 in these experiments, whereas CHMP4A and CHMP4B were not (*Figure 7—figure supplement 1A*). ULK3 phosphorylation was retained in a CHMP4C deletion mutant that lacked both the ALIX-binding site (*McCullough et al., 2008*) and the C-terminal insertion that is phosphorylated by Aurora B[23,24] (CHMP4C 1–200; *Figure 7—figure supplement 1B*, lane 3). However, ULK3 phosphorylation was lost upon further deletion of the MIM2 element (CHMP4C 1–184; *Figure 7—figure supplement 1B*, lane 4). These results show that ULK3 and Aurora B target different residues in CHMP4C and indicate that ULK3-dependent phosphorylation requires the MIM2 region. In agreement with the in vitro data, CHMP4C phosphorylation was also observed in cells co-transfected with Myc-CHMP4C and OSF-ULK3 (*Figure 7—figure supplement 1C*) as was observed for other ULK3 ESCRT-III substrates.

We have previously shown that CHMP4C is phosphorylated at the start of mitosis and dephosphorylated as mitosis progresses (*Carlton et al., 2012*). This phosphorylation induces a mobility shift that can be followed by Western blot of cells expressing HA-CHMP4C at functional

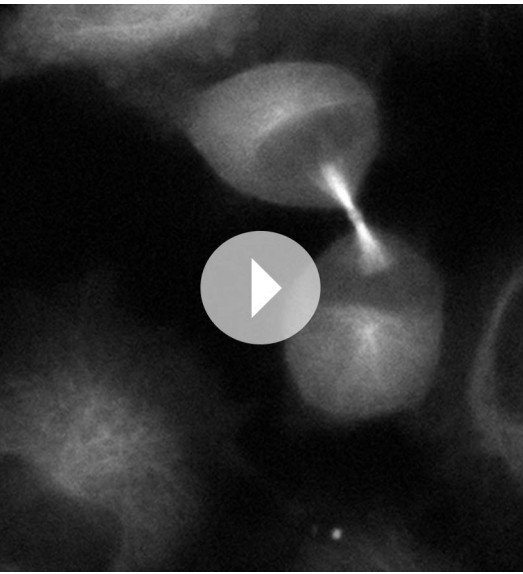

**Video 18.** Representative example of asynchronous HeLa cells stably expressing YFP-Tubulin and empty vector transfected with NT siRNA. Midbody resolution is indicated with an arrow. Abscission time is 90 min. Related to *Figure 6*.

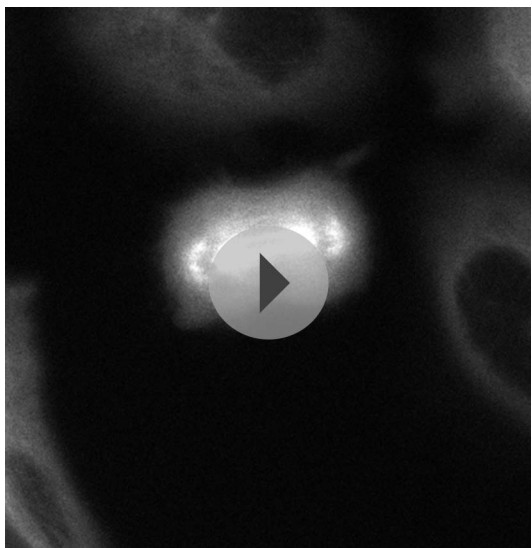

**Video 19.** Representative example of asynchronous HeLa cells stably expressing YFP-Tubulin and empty vector transfected with IST1 siRNA. Midbody resolution is indicated with an arrow. Abscission time is 190 min. Related to *Figure 6*.

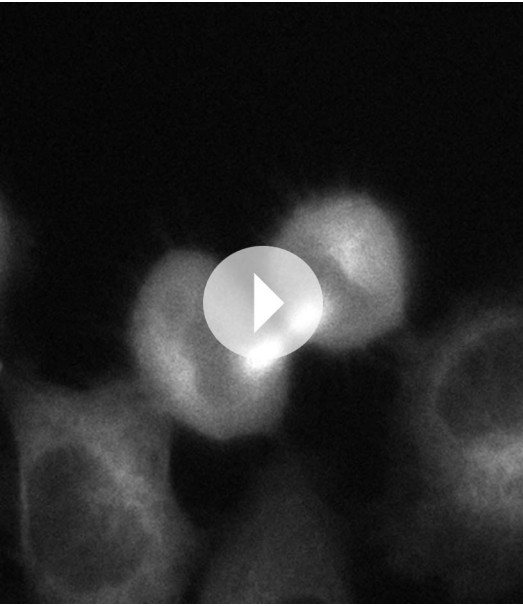

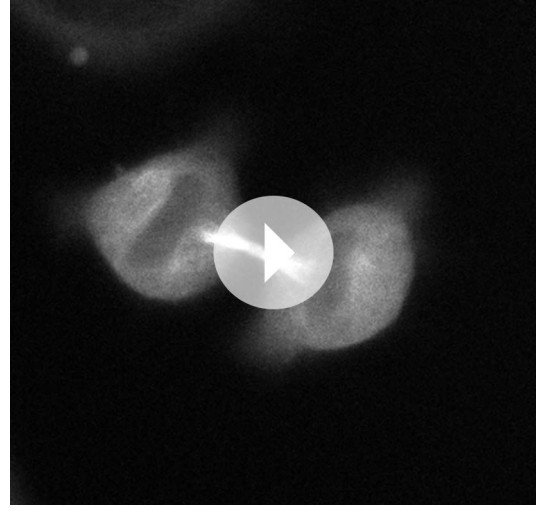

**Video 20.** Representative example of asynchronous HeLa cells stably expressing YFP-Tubulin and siRNA-resistant IST1 (IST1R) WT transfected with IST1 siRNA. Midbody resolution is indicated with an arrow. Abscission time is 90 min. Related to **Figure 6**.

**Video 21.** Representative example of asynchronous HeLa cells stably expressing YFP-Tubulin and IST1R 4SA transfected with IST1 siRNA. Midbody resolution is indicated with an arrow. Abscission time is 90 min. Related to **Figure 6**.

levels (**Carlton et al., 2012**) (**Figure 7—figure supplement 1D**). Importantly, mitotic CHMP4C phosphorylation was also observed for a HA-CHMP4C mutant that lacked the Aurora B target residues (CHMP4CδINS) (**Figure 7—figure supplement 1D**). Thus, Aurora B-independent phosphorylation of CHMP4C occurs during cell division. Quantitative Western blot revealed that mitotic arrest increased the fraction of phosphorylated CHMP4C (P-CHMP4C) to 81% vs 7% in asynchronous cells (**Figure 7H**, lane 3 vs 1).

However, depletion of ULK3 reduced the mitotic fraction of P-CHMP4C to just 30% (**Figure 7H**, lane 4). These differences could not be attributed to gross changes in mitotic arrest because P-H3 levels remained constant upon ULK3 depletion. Similarly, the fraction of phosphorylated CHMP4CδINS increased in mitotic cells (**Figure 7H**, lane 7 vs 3), and phosphorylation levels were again reduced by ULK3 depletion, albeit not as dramatically as in the case of WT CHMP4C (**Figure 7H**, lane 8 vs 7 and 4). Hence, ULK3 contributes to mitotic CHMP4C phosphorylation, and ULK3-dependent phosphorylation and Aurora B-dependent phosphorylation of CHMP4C are separate events.

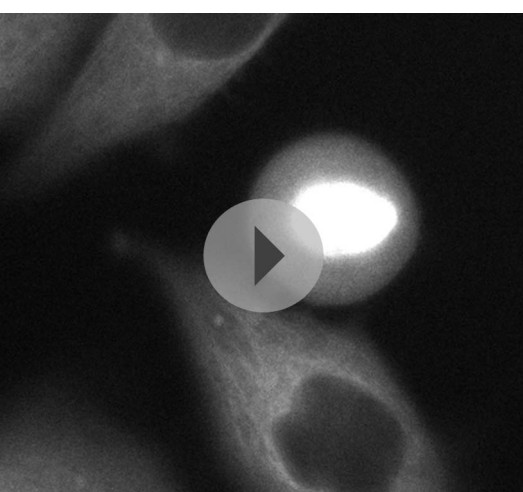

**Video 22.** Representative example of asynchronous HeLa cells stably expressing YFP-Tubulin and IST1R 4SE transfected with IST1 siRNA. Midbody resolution is indicated with an arrow. Abscission time is 180 min. Related to **Figure 6**.

## Discussion

Our study reveals that ULK3 is an essential component of the abscission checkpoint. ULK3 provides inhibitory signals to the abscission machinery that are sustained in response to defective nuclear pore complex assembly or the presence of lagging chromosomes within the midbody, thereby ensuring faithful cytokinesis

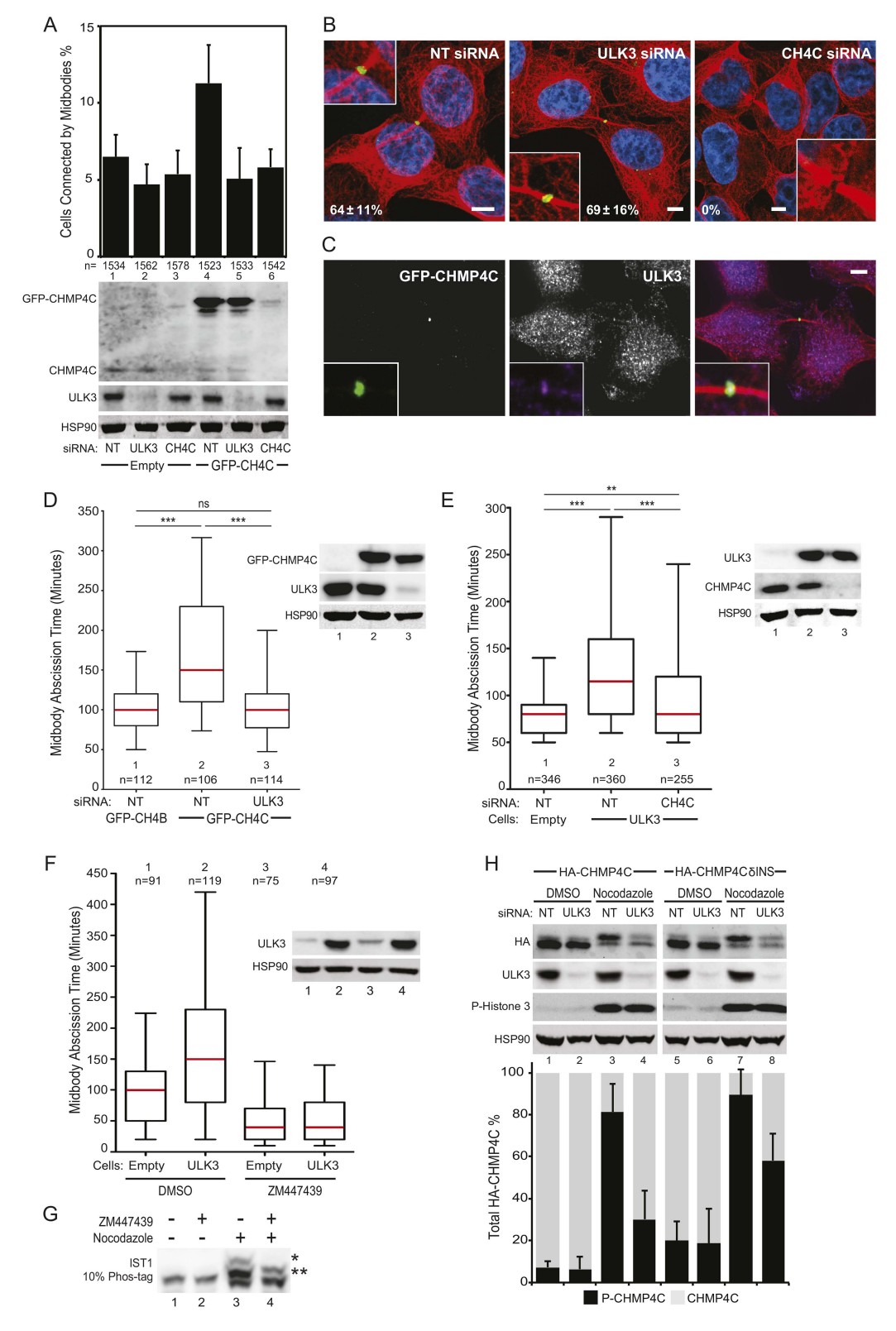

**Figure 7**. ULK3 and CHMP4C are functionally interconnected within the abscission control pathway. (**A**) HeLa cells expressing GFP-CHMP4C were transfected with NT, ULK3, or CHMP4C siRNA, fixed and stained with Hoechst and α-Tubulin antibody to visualize multinucleated and cells connected by midbodies. Data are represented as mean percentage of midbody-arrested cells ±SD from three separate experiments. (**B**) Confocal microscopy of HeLa

*Figure 7. continued on next page*

*Figure 7. Continued*

GFP-CHMP4C cells treated as in (**A**). Magnifications of the midbody are shown. Data are represented as mean percentage of CHMP4C positive midbodies ±SD from two separate experiments (NT n = 110; ULK3 n = 43; CHMP4C n = 53). Bars = 5 µm. (**C**) HeLa GFP-CHMP4C cells were stained with α-ULK3 antibody. Panel shows a representative example of GFP-CHMP4C at the Flemming body co-localizing with endogenous ULK3 (94% of cases, n = 31). Merged channels are shown on the right (red: Tubulin, green: GFP, purple: ULK3, blue: nuclei). Magnifications of the midbody are shown for each channel. Bar = 5 µm. (**D**) HeLa GFP-CHMP4B or GFP-CHMP4C expressing cells were treated as in (**A**) and imaged live. Midbody resolution times were quantified in three separate experiments (mean times were 1: 106 ± 52 min; 2: 180 ± 114 min; 3: 105 ± 47 min). (**E**) HeLa cells expressing empty vector or ULK3 WT were transfected with the indicated siRNA, and midbody resolution times were scored in three separate experiments (mean times were 1: 84 ± 34 min; 2: 136 ± 87 min; 3: 110 ± 79 min). n = total number of events analyzed per sample. (**F**) HeLa cells expressing empty vector or ULK3 WT were treated with 1 µM DMSO or Aurora B inhibitor (ZM447439). Only cells at midbody stage were monitored and imaged live starting at the time of treatment. The time spent in abscission until midbody resolution was analyzed in two separate experiments (mean times were 1: 100 ± 57 min: 2: 171 ± 122 min: 3: 53 ± 41 min: 4: 54 ± 41 min). (**G**) HeLa cells were treated overnight with media containing DMSO or Nocodazole with or without 1 µM ZM44739 inhibitor. Cell lysates were resolved on 10% Phos-tag gels and analyzed by Western blot. 'Asterisks' denote bands corresponding to IST1 phosphorylated species that were sensitive (*) or insensitive (**) to Aurora B inhibition. (**H**) HeLa cells stably expressing HA-CHMP4C^R or HA-CHMP4C^RδINS (lacks residues 201–217) were transfected with NT or ULK3 siRNA prior to overnight treatment with media containing DMSO or Nocodazole. HA-CHMP4CR WT or δINS levels were quantified by infrared imaging; data are represented as mean percentage of phosphorylated CHMP4C (P-CHMP4C, higher molecular weight band) vs non-phosphorylated CHMP4C (lower molecular weight) ±SD from three separate experiments. Cell lysates in (**A**, **D**-**F**) were analyzed by Western blot with the indicated antibodies. See also *Figure 7—figure supplement 1* and *Videos 23–28*.

The following figure supplement is available for figure 7:

**Figure supplement 1**. ULK3 phosphorylates CHMP4C.

progression. ULK3 is also the first identified component of a pathway that delays abscission in response to midbody tension.

Both ULK3 kinase and ESCRT-III-binding activities are required for abscission regulation and our data support a model in which ULK3 phosphorylates ESCRT-III proteins, and thereby, delays the membrane cleavage step of abscission. Consistent with this model, we have identified a subset of ESCRT-III proteins, particularly IST1, which are bound and phosphorylated by ULK3. The MIT2 domain of ULK3 binds the MIM1 of IST1 with unusually high affinity. In contrast, both MIT domains contribute significantly to CHMP1A/B binding, suggesting that MIM1 elements of CHMP1 and IST1 proteins bind ULK3 in distinct ways. Both MIT domains contribute to ULK3 functions in the abscission checkpoint. These observations suggest that IST1

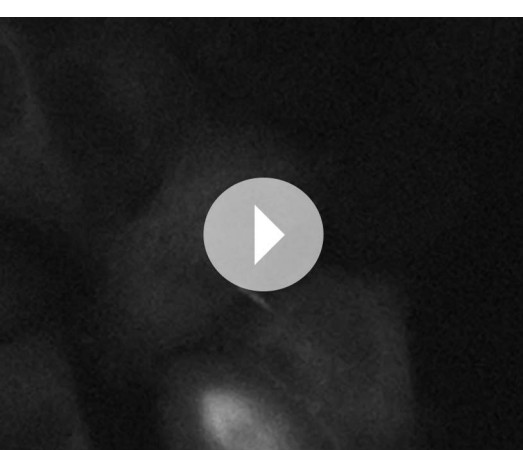

**Video 23.** Representative example of asynchronous HeLa cells stably expressing GFP-CHMP4B and mCherry-Tubulin transfected with NT siRNA. Midbody resolution is indicated with an arrow. Abscission time is 110 min. Related to *Figure 7*.

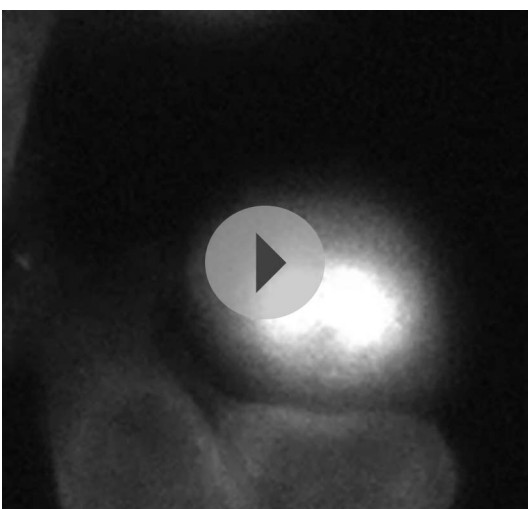

**Video 24.** Representative example of asynchronous HeLa cells stably expressing GFP-CHMP4C and mCherry-Tubulin transfected with NT siRNA. Midbody resolution is indicated with an arrow. Abscission time is 140 min. Related to *Figure 7*.

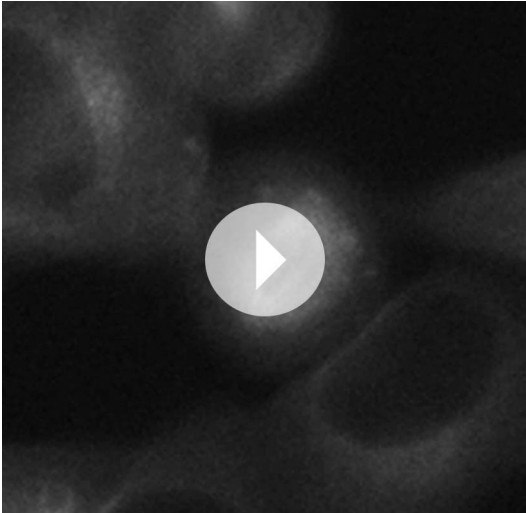

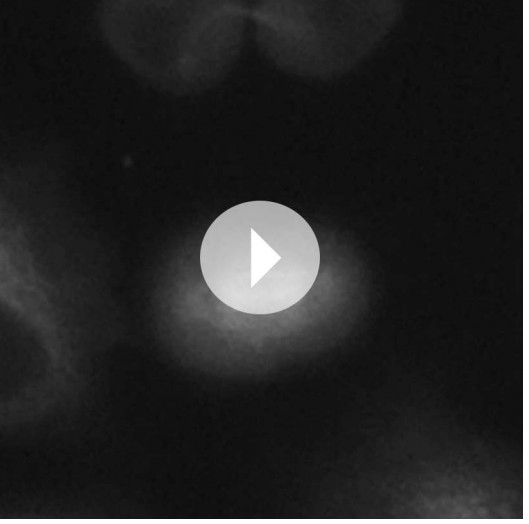

**Video 25.** Representative example of asynchronous HeLa cells stably expressing GFP-CHMP4C and mCherry-Tubulin transfected with ULK3 siRNA. Midbody resolution is indicated with an arrow. Abscission time is 80 min. Related to *Figure 7*.

**Video 26.** Representative example of asynchronous HeLa cells stably expressing mCherry-Tubulin and empty vector transfected with NT siRNA. Midbody resolution is indicated with an arrow. Abscission time is 80 min. Related to *Figure 7*.

engages the ULK3 MIT2 domain, whereas the other MIT site is occupied by other ESCRT-III proteins and/or by yet unidentified binding partners.

IST1 phosphorylation is required to mediate the abscission checkpoint and an IST1 phosphomimetic mutant failed to support abscission, indicating that phosphorylation inhibits at least one IST1 abscission function. We can envision different, non-exclusive mechanisms by which ULK3 phosphorylation of ESCRT-III proteins could delay

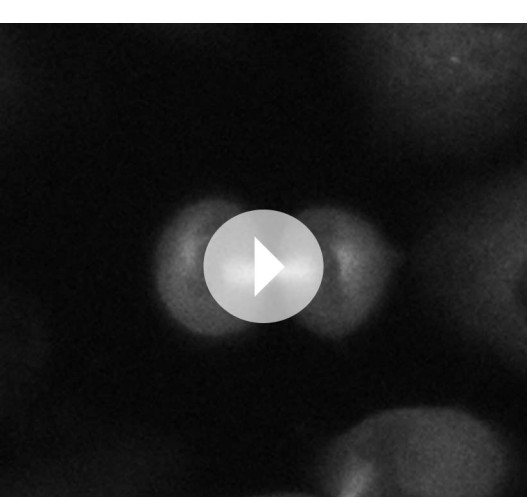

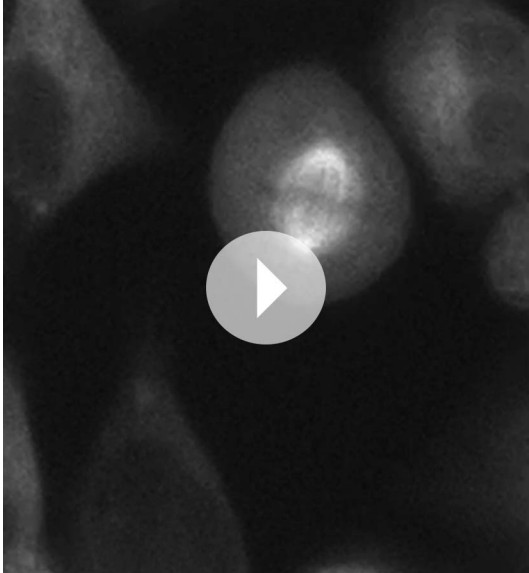

**Video 27.** Representative example of asynchronous HeLa cells stably expressing mCherry-Tubulin and ULK3 transfected with NT siRNA. Midbody resolution is indicated with an arrow. Abscission time is 120 min. Related to *Figure 7*.

**Video 28.** Representative example of asynchronous HeLa cells stably expressing mCherry-Tubulin and ULK3 transfected with CHMP4C siRNA. Midbody resolution is indicated with an arrow. Abscission time is 90 min. Related to *Figure 7*.

abscission. In one model, ULK3 phosphorylation could directly prevent extension of ESCRT-III filaments to the abscission site either because phosphorylation stabilizes the auto-inhibited ESCRT-III conformation (*Muziol et al., 2006*; *Lata et al., 2008*; *Bajorek et al., 2009b*) or inhibits essential interactions within the polymer.

ESCRT-III phosphorylation could also modulate interactions with other components of the abscission machinery. Overexpression of catalytically active ULK3 induced both IST1 and CHMP4B to form single rings within the Flemming body, a phenotype rarely observed in control cells. Intriguingly, both endogenous ULK3 and functionally active CHMP4C also localize to the Flemming body (*Capalbo et al., 2012*; *Carlton et al., 2012*), suggesting that ULK3 may promote association of ESCRT-III subunits with abscission checkpoint regulators to form a cytokinetic inhibitory complex at the central region of the midbody. We also observed that phosphomimetic mutations enhanced IST1 interactions with the late-acting VPS4 protein and its activator LIP5 in cells. We speculate that these enhanced interactions are mediated by unidentified bridging factor(s) because the IST1 phosphorylation sites lie outside of the VPS4- and LIP5-binding sites. The Abscission/NoCut Checkpoint Regulator (ANCHR) protein delays abscission by retaining VPS4 at the Flemming body (*Thoresen et al., 2014*), and it is, therefore, possible that phosphorylated IST1 may collaborate with ANCHR. Moreover, IST1 can also inhibit VPS4 ATPase activity by forming an inactive IST1-VPS4 heterodimer (*Dimaano et al., 2008*), and phosphorylated IST1 could, therefore, delay abscission by inhibiting VPS4 ATPase activity or by inhibiting productive VPS4 recruitment by ESCRT-III proteins, including CHMP1 and IST1, which help localize VPS4 to the midbody late in cytokinesis (*Agromayor et al., 2009*; *Bajorek et al., 2009a*).

We have shown that ULK3 acts in concert with Aurora B to regulate abscission through CHMP4C. We favor a model in which ULK3 is activated during mitosis and functions downstream of Aurora B and CHMP4C. This would explain why mitotic phosphorylation of IST1 requires ULK3 and CHMP4C and why ULK3 overexpression can partially overcome the requirement for CHMP4C in abscission delay. Given that ULK3 phosphorylates multiple ESCRT-III subunits, the initial phosphorylation of CHMP4C by Aurora B could be subsequently 'amplified' by ULK3. Our data further show that ULK3 and Aurora B phosphorylate CHMP4C at different sites (*Carlton et al., 2012*). This is consistent with the functional interdependence of ULK3 and CHMP4C in the abscission checkpoint and could contribute to creating a feedback loop that sustains the inhibitory signal. Mitotic phosphorylation of CHMP4C then decreases at (or near) the time of abscission, suggesting that a checkpoint-dependent phosphatase may override ULK3 activity and provide an 'all clear' signal to complete abscission.

Finally, the recent identification of CHMP4C variants as risk factors for ovarian cancer suggests that abscission checkpoint defects may increase the tumorigenic potential of the daughter cells (*Pharoah et al., 2013*). The identification of ULK3 as an essential component of the abscission checkpoint, thus, opens a new avenue in the emerging relationship between the abscission checkpoint and tumor formation.

## Materials and methods

Lists of plasmids, siRNA, antibodies, and cell lines used in this study are provided in *Supplementary file 1*.

### Y2H assay

Yeast Y190 cells were co-transformed with 1 µg of each plasmid encoding the indicated proteins fused to the VP16 activation domain (pHB18) or the Gal4 DNA-binding domain (pGBKT7). Co-transformants were selected on SD-Leu-Trp agar for 3 days at 30°C. Protein–protein interactions were determined by β-galactosidase activity in yeast extracts using chlorophenol red-β-D-galactopyranoside (Roche, Switzerland) as a substrate.

### Cell culture

HeLa and 293T cells were cultured in Dulbecco's Modified Eagle Medium (DMEM) containing 10% Fetal Calf serum (FCS) and gentamycin 20 µg/ml; (no antibiotics were used to culture cells for experiments shown in *Figures 1A,C,5B*, *Figure 5—figure supplement 1* and *Figure 7—figure supplement 1C*). Stable cells lines were generated using MLV-based retroviruses as described previously (*Carlton and Martin-Serrano, 2007*) and selected using puromycin (200 ng/ml) or G418 (500 µg/ml). For live-cell imaging experiments, imaging was performed between 4 and 10 days after selection was added to the transduced cells.

## Transfections

Polyethylenimine (PEI; Polysciences, Germany) or Lipofectamine 2000 (Life Technologies, Carlsbad, CA) was used to transfect plasmids into 293T and HeLa cells respectively, according to manufacturer's instructions. For all siRNA assays, cells were transfected twice with 100 nM of siRNA using Dharmafect-1 (Dharmacon RNA Technologies, Lafayette, CO). For IST1 rescue experiments in *Figure 6A,B*, cells were transfected once with 100 nM of siRNA and fixed/imaged 12–16 hr after transfection. To trigger the abscission pathway by using partial depletion of NUP153, cells were co-transfected with 10 nM of NUP153 siRNA (diluted in NT siRNA).

## Genome editing with CRISPR-Cas9 system

Two sets of guide RNA targeting the ULK3 locus were designed using the Zhang Lab website (http://crispr.mit.edu) and cloned into a lentiCRISPRv2 plasmid (Addgene, Cambridge, MA) (*Sanjana et al., 2014*). The sequences for the two sets of guide oligos were (quoted 5′ to 3′) set 1: CACCGCACGTACGCCACGGTGTACA and AAACTGTACACCGTGGCGTACGTGC; set 2: CACCGGGATCTCAATCTCCGTGAGG and AAACCCTCACGGAGATTGAGATCCC. Stable cell lines were generated by transfecting a 10-cm dish of 293T cells per construct, using 7.2 µg of lentiCRISPR, 7.2 µg p8.91 HIV Gag-pol, and 1.8 µg of VSV-G. 48 hr post-transfection, released virions were collected and concentrated through a 20% sucrose gradient by ultracentrifugation at 28,000 rpm for 1.15 hr at 4°C. Pelleted virions were re-suspended in serum free media overnight and used to transduce HeLa cells in 6-well per plate by spinoculation for 2 hr. After 2 days, puromycin (100 ng/ml) was used for selection and cells were maintained under selection until single cell clones were obtained by limiting dilution. After confirmation of efficient ULK3 knockout by Western blot, two δULK3 clones were selected and used for functional experiments.

## OSF-ULK3 pull downs

For the experiment shown in *Figures 1A*, $2 \times 10^6$ 293T cells were seeded in 10-cm dishes and each dish was singly transfected 18–24 hr later with plasmids encoding Myc-tagged ESCRT-III proteins or OSF-ULK3 (6 µg/dish) using PEI. To equalize expression levels, the following amounts of plasmids encoding Myc-tagged ESCRT-III proteins were transfected (with empty vector used as necessary to bring the total to 10 µg): 10 µg each of CHMP2A-Myc, CHMP2B-Myc, CHMP4B-Myc, CHMP4C-Myc, and CHMP6-Myc; 5 µg of CHMP4A-Myc; 1.5 µg of CHMP1A-Myc and Myc-IST1; 0.25 µg of CHMP1B-Myc, CHMP3-Myc, CHMP5-Myc, and CHMP7-Myc. Cells were harvested 48 hr post-transfection and lysed in 50 mM Tris pH 7.2, 150 mM NaCl, 0.5 mM $MgCl_2$, 5 mM β-mercaptoethanol (BME), 0.2% Triton X-100 supplemented with DNAse I (Roche), and protease inhibitors (Sigma–Aldrich, St. Louis, MO). Lysates were clarified by centrifugation at $16,100 \times g$ for 10 min at 4°C. Lysates expressing ESCRT-III proteins were further diluted with lysate from untransfected cells to match expression levels and mixed with lysates prepared from cells expressing OSF-ULK3. Mixed lysates were incubated overnight (~18 hr) at 4°C with 20 µl of pre-equilibrated streptactin sepharose resin (IBA Lifesciences, Germany) and washed once with wash buffer: 50 mM Tris, pH 7.2, 350 mM NaCl, 5 mM BME, 0.5 mM $MgCl_2$, 0.2% Triton X-100 and three times with wash buffer containing 150 mM NaCl. After the final wash, the streptactin beads were aspirated to near dryness, and bound proteins were eluted by boiling in 20 µl of Laemmli sample buffer, resolved by SDS-PAGE, and examined by Western blotting.

For the experiment shown in *Figure 1C*, 293T cells were seeded in a 6-well plate at $0.25 \times 10^4$ cells/well and co-transfected 18–24 hr later with 1.5 µg of the indicated plasmids. Cells were harvested 48 hr post-transfection, lysed, and clarified as described above. Cell lysates were incubated with 20 µl of streptactin resin for 2 hr at 4°C. The streptactin resin was washed four times in lysis buffer and aspirated to near dryness. Bound proteins were eluted by boiling in 20 µl of Laemmli sample buffer, resolved by SDS-PAGE, and examined by Western blotting.

## GST pull downs

For co-precipitation experiments, 293T cells from 6-well plates were transfected with 1 µg of plasmids encoding the indicated pCAGGS/GST and pCR3.1-YFP fusion proteins. 48 hr later, cells were lysed in 1 ml of lysis buffer (50 mM Tris, pH 7.4, 150 mM NaCl, 5 mM EDTA, 5% glycerol, 1% Triton X-100 and a protease inhibitor cocktail [complete mini-EDTA free, Roche]). Clarified lysates were incubated with

glutathione–sepharose beads (Amersham Biosciences, Pittsburgh, PA) for 3 hr at 4°C and washed three times with wash buffer (50 mM Tris, pH 7.4, 150 mM NaCl, 5 mM EDTA, 5% glycerol, and 0.1% Triton X-100). Bound proteins were eluted by boiling in 100 µl of Laemmli sample buffer, resolved by SDS-PAGE, and examined by Western blotting.

## Immunoprecipitations

$4.5 \times 10^6$ 293T cells were transfected with 20 µg of plasmids encoding HA-tagged proteins. After 48 hr, cells were lysed in 5 ml of lysis buffer (50 mM Tris, pH 7.4, 150 mM NaCl, 5 mM EDTA, 5% glycerol, 1% Triton X-100) containing a protease inhibitor cocktail (complete mini-EDTA free, Roche) and a phosphatase inhibitor mixture (PhosStop, Roche). After sonication, cleared lysates were incubated with 5 µg of monoclonal α-HA for 1 hr at 4°C, and then 40 µl of protein G-agarose for 2 hr at 4°C. Bead-bound complexes were washed four times in wash buffer (50 mM Tris, pH 7.4, 150 mM NaCl, 5 mM EDTA, 5% glycerol, and 0.1% Triton X-100), and bead-bound proteins were eluted by boiling in 80 µl of sample buffer, and examined by SDS-PAGE and Western blot.

## Proximity-dependent biotin identification

This approach is based on the fusion of ULK3 to the *Escherichia coli* modified biotin protein ligase BirA-113G, which can promiscuously biotinylate vicinal proteins in vivo that are later isolated with streptavidin-coated beads (*Roux et al., 2012*). For these experiments, $2.5 \times 10^6$ HeLa cells stably expressing BirA-ULK3 were seeded in 10-cm dishes and treated overnight with 100 µM Biotin (Sigma–Aldrich). Cells were dissociated with Phosphate Buffer Saline (PBS) 0.5 mM EDTA, washed once with 1X PBS, and lysed in 500 µl of lysis buffer (50 mM Tris, pH 7.4, 500 mM NaCl, 5 mM EDTA, 0.4% SDS, 1 mM DTT and protease inhibitor cocktail). After a first pulse of sonication, Triton X-100 was added to a final concentration of 2%, samples were briefly sonicated again, and an equal volume of cold 50 mM Tris (pH 7.4) was added prior to a final pulse of sonication. Lysates were cleared by centrifugation for 5 min and incubated with avidin–agarose beads (Pierce, Rockford, IL) for 3 hr at 4°C and washed three times at 25°C with lysis buffer. Bound proteins were eluted by boiling in 100 µl of Laemmli sample buffer, resolved by SDS-PAGE, and examined by Western blotting.

## Western blotting

Cell lysates or bead eluates were denatured in Laemmli buffer and resolved on polyacrylamide. Where indicated, Phos-tag (Wako Chemicals, Japan) and $MnCl_2$ were added into 10% gels to induce mobility shifts in phosphorylated proteins, to final concentrations of 40 µM and 80 µM, respectively, for *Figure 5B*, *Figure 5—figure supplement 1A* and *Figure 7—figure supplement 1C*, or 25 µM and 50 µM, respectively, for *Figures 5D,E,7G*. Proteins were transferred onto nitrocellulose or PVDF membranes and probed with the indicated antibody in 1% or 5% milk. IRDye-conjugated secondary antibodies (Li-cor Biosciences, Lincoln, NE) were imaged using an Odyssey infrared scanner (Li-cor Biosciences). HRP-conjugated secondary antibodies (Pierce) were visualized using Amersham ECL Prime Western blotting detection reagent (GE Healthcare, Pittsburgh, PA) and ImageQuant LAS4000 system (GE Healthcare).

## In vitro kinase assays

HA-tagged proteins were immunoprecipitated as described above. After four washes in wash buffer, bead-bound complexes were washed twice in wash buffer containing 0.5 M NaCl and twice in kinase assay buffer (50 mM Tris, pH 7.4, 150 mM NaCl, 25 mM KCl, 5 mM $MgCl_2$, 0.02% Triton X-100). Beads were resuspended in 40 µl of kinase assay buffer supplemented with 1 mM DTT. 100 ng of recombinant His-ULK3 (Abcam, UK) was added on ice where required. 2.5 µCi of ATP γ-$^{32}$P was added and reactions were incubated with mixing at 30°C for 30 min. Reactions were terminated by addition of Laemmli buffer and boiling for 5 min. Lysates were then resolved by SDS-PAGE and transferred to nitrocellulose. Membranes were exposed to a storage phosphor screen, visualized using a Typhoon 9400 (GE Healthcare) and processed using ImageQuant. Subsequent Western blotting with polyclonal α-HA allowed visualization of immunoprecipitated proteins.

Alternatively, when using ULK3 GST-fusion plasmids, $2.2 \times 10^6$ 293T cells were co-transfected with 2 µg of the indicated GST plasmid. After 48 hr, cells were harvested and lysed in 2 ml of GST

pull-down lysis buffer. Cleared lysates were incubated with 25 μl glutathione–sepharose beads (Amersham Biosciences) for 2 hr at 4°C and washed twice with wash buffer, twice with wash buffer/0.5 M NaCl, and twice with kinase assay buffer. Kinase assays proceeded as described above.

## Expression and purification of recombinant proteins

IST1 (residues 316–366, expressed as a His-SUMO-fusion protein) and GST-fusions of ULK3 (MIT)$_2$ (residues 277–449) (WT and V338D or M434D mutants) and ULK3 MIT2 (359–449) were each expressed in 1 L cultures of BL21-Codon Plus (DE3) RIPL cells (Agilent, Santa Clara, CA) in ZYP-5052 auto-induction media (*Studier, 2005*). For crystallographic studies, ULK3(MIT)$_2$ was additionally expressed in 2 L auto-induction PA-5052 media containing selenomethionine (SeMet) (*Studier, 2005*). All purification steps were performed at 4°C. For GST-fusions, cells were resuspended in lysis buffer (40 ml/L of culture) containing 50 mM Tris pH 8.0, 150 mM NaCl, 0.5 mM EDTA, 2 mM DTT, 0.125% sodium deoxycholate supplemented with protease inhibitors (aprotinin, leupeptin, pepstatin, and PMSF), lysozyme, and DNAse I (Roche). Cells were lysed by sonication, and the cell lysate was clarified by centrifugation for 45 min at 32,000×*g*. Clarified lysate was incubated for 1–2 hr with 15 ml GST resin (GE Healthcare) equilibrated with binding buffer: 50 mM Tris, pH 8.0, 150 mM NaCl, 2 mM DTT, and washed with 10 column volumes of binding buffer containing 500 mM NaCl and 10 column volumes of 150 mM NaCl-containing binding buffer. GST-affinity tags were removed from resin-bound ULK3 proteins by overnight incubation with PreScission Protease (0.3 mg) in 40 ml of GST-binding buffer at 4°C. Cleaved proteins were collected from the column flow through and dialyzed against Q-sepharose-binding buffer (20 mM Tris, pH 8.0, 50 mM NaCl, 2 mM DTT, 0.5 mM EDTA) for further purification by Q-sepharose anion exchange chromatography (GE Healthcare Life Sciences) with a linear gradient from 50 to 500 mM NaCl. Fractions containing ULK3 were pooled and dialyzed against gel filtration buffer (25 mM Tris, pH 7.2, 150 mM NaCl, 2 mM DTT, 0.5 mM EDTA) and further purified by Superdex-75 size exclusion chromatography (GE Healthcare Life Sciences). Typical protein yields were ~15–20 mg/L culture in ZYP-5052 (rich) media and 2.5 mg/L culture in PA-5052 SeMet media.

Cells expressing His-SUMO-IST1$_{316-366}$ were lysed by sonication in buffer (40 ml/L of culture) containing 50 mM Tris, pH 7.2, 200 mM NaCl, 5 mM imidazole, 5 mM DTT, 0.5 mM EDTA, and 0.125% sodium deoxycholate, supplemented with lysozyme, protease inhibitors, and DNAse I (Roche). Clarified cell lysate was incubated with 10 ml of cOmplete His-Tag purification resin (Roche) for 20 min, washed with 10 column volumes of wash buffer: 50 mM Tris, pH 7.2, 500 mM NaCl, 5 mM Imidazole, 5 mM DTT, 0.5 mM EDTA followed by 10 column volumes of wash buffer containing 150 mM NaCl. The His-SUMO affinity tag was removed from resin bound IST1 peptide by overnight incubation with UPL1 protease (0.1 mg) in 40 ml of the 150 mM NaCl wash buffer at 4°C. The cleaved IST1 peptide was collected from the column flow through and further purified by anion exchange and gel filtration chromatography as described above for ULK3 proteins. Typical IST1 peptide yields were 4.5 mg/L culture. Purified IST1 and ULK3 proteins contain non-native 'GC' and 'GPHM' residues at their N-termini, respectively. Masses of purified proteins were confirmed by ESI/MS as follows: IST1$_{313-366}$ calculated = 5733 Da and experimental = 5732 Da; ULK3 (MIT)$_2$ calculated = 19,355 Da and experimental 19,355 Da; ULK3(MIT)$_2$ M434D calculated = 19,399 Da and experimental = 19,399 Da; ULK3(MIT)$_2$ V338D calculated = 19,371 Da and experimental = 19,371 Da; ULK3 MIT2 calculated = 10,535 Da, and experimental = 10,535 Da.

Uniformly enriched $^{13}$C,$^{15}$N-IST1$_{303-366}$ was expressed as a TEV-cleavable GST fusion protein in 1 L of M9 minimal medium containing 2 g each of $^{15}$N-ammonium chloride and $^{13}$C-glucose. Cells were lysed by sonication in 40 ml of lysis buffer: 50 mM Tris pH 8.0, 100 mM NaCl, 1 mM DTT, 0.125% sodium deoxycholate, 0.1% Tween-20 in the presence of protease inhibitors (aprotinin, leupeptin, pepstatin, and PMSF) and lysozyme. Lysates were clarified as described above, and the supernatant was incubated with ~15 ml of GST-sepharose resin (GE Healthcare Life Sciences) overnight at 4°C. The resin was washed with ~10 column volumes of 20 mM Tris, pH 8.0, 100 mM NaCl, 1 mM DTT, and bound protein was eluted in wash buffer containing 20 mM reduced *l*-glutathione. Eluted $^{15}$N,$^{13}$C-IST1$_{303-366}$ was dialyzed against 20 mM Tris, pH 8.0, 100 mM NaCl, 1 mM DTT in the presence of TEV protease (at room temperature) to remove the GST tag. Following TEV cleavage, the IST1 peptide was dialyzed into Q-sepharose buffer: 20 mM Tris pH 8.0, 50 mM NaCl, 0.5 mM DTT, and purified by ion-exchange chromatography over a Q-sepharose column (GE Healthcare Life Sciences) using a linear gradient of 50–500 mM NaCl over ~18 column volumes. Fractions containing $^{15}$N,$^{13}$C-IST1$_{303-366}$ were exchanged into 20 mM sodium phosphate pH 6.2, 25 mM

NaCl, 0.1 mM EDTA, 0.1 mM NaN$_3$, 0.5 mM DTT, and 10% D$_2$O for NMR spectroscopy. The yield of purified $^{15}$N,$^{13}$C-IST1$_{303-366}$ peptide was ~1.4 mg/L cell culture. The expected protein mass was confirmed by ESI/MS of an unlabeled sample: calculated = 7217 Da and experimental = 7216 Da.

## NMR spectroscopy and data analysis

NMR data were recorded on a Varian INOVA 600 MHz spectrometer equipped with a cryogenic probe, processed using FELIX 2007 (Felix NMR, Inc.), and analyzed using SPARKY3 (*Goddard and Kneller, 2001*), AutoAssign (*Zimmerman et al., 1997*), and NMRViewJ (OneMoon Scientific). IST1$_{303-366}$ backbone resonances were assigned using standard triple resonance experiments (*Cavanagh, 2007*) from a sample containing 0.4 mM uniformly $^{13}$C- and $^{15}$N-enriched IST1 in 20 mM sodium phosphate pH 6.2, 25 mM NaCl, 0.1 mM EDTA, 0.1 mM NaN$_3$, 0.5 mM DTT, and 10% D$_2$O. The chemical shifts for IST1 have been deposited in the Biological Magnetic Resonance Bank under accession number 25393.

Chemical shift perturbations induced by ULK3 binding to IST1 were identified by titrating 0.4 mM uniformly $^{13}$C- and $^{15}$N-enriched IST1 with increasing amounts of unlabeled ULK3 to final stoichiometries of (IST1 : ULK3): 0, 0.25, 0.5, 1.0, 1.1, and 2.0. The complex was in slow exchange on the NMR time scale. Chemical shifts were analyzed in NMRViewJ (OneMoon Scientific), and IST1 resonances that were shifted more than ½ peak width in the bound complex were given a score of '+1'. Unperturbed resonances were scored as '−1' and resonances that could not be unambiguously assigned to either category owing to spectral overlap were scored as '0'.

## Fluorescence polarization

IST1 MIMs peptide (residues 316–366, containing a non-native N-terminal 'GC' dipeptide) was expressed and purified as a His-SUMO fusion protein as described above. MIM1 (residues 344–366, containing a non-native, N-terminal cysteine) and MIM2 (residues 316–343, containing a non-native N-terminal 'GC' dipeptide) peptides were synthesized, purified, and fluorescently labeled in the University of Utah Peptide Synthesis Core. Briefly, MIM1 and MIM2 peptides were synthesized on an ABI 433 synthesizer (Applied Biosystems, Waltham, MA) with a cysteine at the N-terminus using Fmoc solid phase technology, common protecting groups, and HBTU chemistry on an ABI 433 synthesizer (Applied Biosystems). Both recombinant and synthesized peptides were purified by reversed phase chromatography using acetonitrile/water gradients with 0.1% trifluoroacetic acid in both solvents. Peptide-containing fractions were pooled and dried, then re-dissolved in DMSO. Fluorescent labeling was performed in DMSO at 4°C with approximately 1.3-fold molar excess of Oregon Green 488 maleimide (Life Technologies/Molecular Probes 06,034) dissolved in a 1:1 solution of acetonitrile: DMSO. The reaction progress was monitored by HPLC, and labeled peptides were separated from free dye and residual unlabeled peptides using the same reversed phase conditions described above. Labeled peptide masses were measured by MALDI-TOF-MS at the University of Utah Mass Spectrometry Core facility: dye-labeled MIMs, calculated = 6196 Da and experimental = 6192 Da; dye-labeled MIM1, calculated = 3298 Da and experimental = 3296 Da; and dye-labeled MIM2, calculated = 3483 Da and experimental = 3480 Da. Confirmed peptide fractions were dried under vacuum, redissolved in water, and concentrations were calculated using the absorbance of Oregon Green 488 at 491 nm (Extinction coefficient 83,000 cm$^{-1}$ M$^{-1}$ in 50 mM potassium phosphate, pH 9). FP experiments were performed in 60-µl binding reactions in binding buffer: 25 mM Tris, pH 7.2, 150 mM NaCl, 0.1 mg/mL Bovine Serum Albumin (BSA), 0.01% Tween-20, and 1 mM Dithiothreitol (DTT) using 250 pM fluor-labeled IST1 peptides and twofold dilutions of ULK3 proteins. FP was measured using a Tecan Infinite 200 plate reader with excitation at 485 nm and detection at 535 nm. Dissociation constants were calculated by fitting the increase in FP to a 1:1 binding equation using KaleidaGraph (Synergy Software) as described previously (*Skalicky et al., 2012*). Each binding isotherm was measured at least three times independently, and mean K$_D$ values are reported ±SD.

## Crystallization and data collection

IST1 protein was mixed in a 1:1 molar ratio with native or SeMet-substituted ULK3 MIT2 protein to a final concentration of 10 mg/ml in 20 mM Tris, pH 8.0, 100 mM NaCl, 2 mM DTT, and 3 mM NaN$_3$. ULK3:IST1 complexes were crystallized at 21°C in sitting drops by mixing equal volumes (2 µl) of protein complex solution and precipitant solution (1.6–1.8 M ammonium sulfate, 0.1 M MES pH

6.4–6.7, 0.01 M CoCl$_2$). Crystals were transferred to a cryoprotectant solution containing precipitant supplemented with 30% glycerol, looped, and flash frozen in liquid nitrogen prior to data collection. X-ray diffraction data were collected remotely (*Soltis et al., 2008*) at SSRL beamlines 11-1 and 12-2. All data were processed using AutoXDS (*Gonzalez and Tsai, 2010*; *Kabsch, 2010a*; *2010b*) at SSRL.

## Structure determination and refinement

Initial attempts to crystallize ULK3 (MIT)$_2$ (residues 277–449) in complex with both IST1 MIMs (residues 316–366) yielded crystals of a proteolytically truncated complex comprising ULK3 MIT2 and IST1 MIM1 fragments. The structure of this complex was determined using a data set derived from a crystal of SeMet-substituted ULK3 (MIT)$_2$ in complex with IST1 MIMs, in which both components had undergone proteolysis. The SeMet ULK3/IST1 complex was solved in space group P3$_1$2$_1$ (a = 82.68 Å, b = 82.68 Å, c = 90.15 Å) with single-wavelength anomalous dispersion data at 0.9791 Å (2.1 Å resolution, *Figure 1—source data 1*). PHENIX (Hybrid Structure Search; HySS) (*Afonine et al., 2012*; *Grosse-Kunstleve and Adams, 2003*; *McCoy et al., 2004*) was used to locate and refine the positions for 3 of 4 possible SeMet sites in ULK3 MIT2. Phases were calculated and used to produce a 3.2 Å electron density map. Models for the three MIT molecules in the asymmetric unit were initially built into the electron density and the ULK3:IST1 models were rebuilt de novo in Coot (*Emsley et al., 2010*) and refined in PHENIX. The refined SeMet model was then used as an initial molecular replacement search model to determine structures from higher resolution native data sets (Phaser in PHENIX). The protein termini were identified in the proteolyzed structure and used to design new constructs (ULK3 MIT2: residues 359–449; IST1 MIM1 residues 344–366) for further crystallization. ULK3 MIT2 was expressed and purified as a recombinant protein (described above), and IST1 MIM1 was synthesized by the University of Utah Peptide Synthesis Core. The complex of the shorter constructs crystallized in space group R32 (a = 79.12 Å, b = 79.12 Å, c = 96.62 Å) with a single ULK3 MIT2: IST1 MIM1 complex in the asymmetric unit and the crystal diffracted to 1.38 Å resolution (*Figure 1—source data 1*). This structure is described herein, and data collection and refinement statistics are provided in *Figure 1—source data 1*. Structure coordinates have been deposited in the RCSB Protein Data Bank under PDB ID 4WZX. Chimera was used to analyze structures and generate figures (*Pettersen et al., 2004*).

## In-cell phosphorylation assays

For experiments shown in *Figure 5B*, *Figure 5—figure supplement 1A*, and *Figure 7—figure supplement 1C*, 293T cells were co-transfected with PEI and plasmids encoding Myc-tagged ESCRT-III proteins and OSF-ULK3 WT, or OSF-ULK3-K44H or empty vector. For IST1, CHMP1A, CHMP1B, and CHMP3 experiments, transfections were performed in a 6-well plate seeded 18–24 hr earlier with $0.25 \times 10^4$ 293T cells/well with the following plasmid amounts: 500 ng of OSF-ULK3 WT, 1.5 µg of OSF-ULK3 K44H, 1.5 µg Myc-IST1, 1 µg CHMP1A-myc, 1 µg CHMP1B-myc, and 1 µg CHMP3-Myc. Each transfection was brought to a total of 3 µg DNA with empty vector. CHMP4C and CHMP2A experiments were performed using cell lysates from 10-cm dishes seeded at $2.2 \times 10^6$ and transfected 18–24 hr later. Transfections contained the following plasmid quantities: 6 µg CHMP4C-Myc, 6 µg CHMP1A-Myc and 3–4 µg OSF-ULK3, 6 µg of OSF-ULK3 K44H, or 6 µg of empty vector. All transfections were brought to a total of 12 µg DNA with empty vector. Cells were harvested 48 hr post-transfection and lysed in buffer containing 50 mM Tris pH 7.2, 150 mM NaCl, 5 mM BME, 0.2% Triton X-100, 10 mM MgCl$_2$ supplemented with DNAse I (Roche) and protease inhibitors (Sigma–Aldrich MOP8340). Cell lysates were clarified as described above. Where noted, 25 µl of each lysate was additionally treated with 40 units of CIP (New England Biolabs, Ipswich, MA) and incubated at 37°C for 1 hr. Gel samples were boiled in Laemmli sample buffer both pre-and post-CIP treatment, resolved by SDS-PAGE, and examined by Western blotting.

## Mass spectrometric analyses of phosphorylated IST1 proteins

Ten 10-cm dishes of 293T cells were co-transfected (PEI) with 6 µg of plasmids expressing OSF-ULK3 or OSF-ULK3 K139R (kinase inactive *Maloverjan et al., 2010*), and 6 µg of Myc-IST1. Transfected cells were harvested after 48 hr and lysed by sonication in 50 mM Tris pH 7.2, 150 mM NaCl, 0.5 mM MgCl$_2$, 5 mM DTT (or BME) in the presence of DNase I (Roche), Phos-Stop (Roche), and mammalian protease inhibitors (Sigma–Aldrich). Cell lysates were clarified by centrifugation at 16,100×$g$ at 4°C for 10 min, and incubated with 75 µl of streptactin resin (IBA-Lifesciences for 2 hr at 4°C. Bound ULK3/IST1 complexes were washed once with lysis buffer containing 500 mM NaCl, three times with lysis buffer

(150 mM NaCl), and eluted with lysis buffer containing 10 mM *d*-desthiobiotin. The OSF-ULK3 K139R/ IST1 elution was additionally incubated with CIP (New England Biolabs, Ipswich, MA) at 37°C for 1 hr. Samples were purified prior to mass spectrometry analysis using a C18 ZipTip (Millipore, Temecula, CA) to remove salts and other small molecule contaminants according to the manufacturer's instructions. Proteins were eluted from the ZipTip with three consecutive 0.75 µl washes of 70% methanol, 1% formic acid, and one 1 µl wash of 98% methanol, 1% formic acid, with 2 µl of 1% formic acid added prior to ESI/MS analysis.

ESI/MS analyses of intact proteins were performed using a Quattro-II mass spectrometer (Micromass, Inc. UK). The ZipTip eluant was infused into the instrument at 3 µl/min flow rate. Data were acquired with a cone voltage of 50 eV, spray voltage of 2.8 kV, and scanning from 600 to 1400 Da in 4 s. Scans were accumulated for ~1 min. Spectra were combined and multiply charged molecular ions were deconvoluted into a molecular mass spectrum using MaxEnt 3.4 software (Micromass, Inc.).

To identify phosphorylation sites, co-purified IST1 and ULK3 proteins were digested in solution with a TPCK-modified trypsin and Lys-C protease mixture (Promega, Madison, WI) or chymotrypsin (Princeton, Adelphia, NJ). Trypsin or chymotrypsin (in 50 mM ammonium bicarbonate) was added to the solution (adjusted to pH 7.9) at a ratio of approximately 1:25 (enzyme: protein). Samples were digested at 37°C overnight with trypsin or for 2–2.5 hr with chymotrypsin.

Immobilized metal affinity chromatography (IMAC) was used to enrich phospho-peptides from the trypsin- and chymotrypsin-digested proteins. The IMAC procedure was performed using SwellGel gallium (III) chelated mini-spin columns (Pierce), with minor modifications to the manufacturer's recommended procedure. Briefly, the trypsin- and chymotrypsin-digested proteins were acidified (pH < 3) with 10% acetic acid. The samples were then incubated on the column for 15 min, washed twice with 50 µl of 0.1% acetic acid, twice with 50 µl of 0.1% acetic acid in 10% acetonitrile, and once with 75 µl of nanopure water. Phosphopeptides were eluted with two 20-µl volumes of 25 mM ammonium bicarbonate (pH 9; adjusted with ammonium hydroxide), followed by 20 µl of 25 mM ammonium bicarbonate in 50% acetonitrile. All three eluant fractions were combined, dried down, and reconstituted in a solution of 5% acetonitrile with 0.1% formic immediately prior to nano-LC-MS/MS analysis.

Both phospho-enriched and un-enriched peptides were analyzed using a nano-LC/MS/MS system equipped with a nano-HPLC pump (2D-ultra, Eksigent, Redwood City, CA) and an ESI-LTQ-FT-ICR mass spectrometer equipped with a nanospray ion source (ThermoElectron Corp., Waltham, MA). Approximately 5 µl of peptide samples were injected onto a dC18 nanobore LC column (75 µm ID × 100 mm length, 3 µm particles, made in house, Atlantis, Waters Corp., Milford, MA). Peptides were separated and eluted over a linear gradient of 5–96% acetonitrile in 0.1% formic acid with a constant total flow rate of 350 nl/min over 78 min. The LTQ-FT-ICR mass spectrometer was operated in the data-dependent acquisition mode with the 'top 10' most intense peaks observed in an FT primary scan selected for on-the-fly peptide fragmentation MS/MS acquisitions in the LTQ linear ion trap portion of the instrument. The LTQ linear ion trap was operated with the following parameters: precursor activation time 30 ms and activation Q at 0.25; collision energy at 35%; dynamic exclusion at low mass of 0.1 Da and high mass at 2.1 Da with one repeat count and 10 s duration. Spectra in the LTQ-FT-ICR were acquired from *m/z* 350 to 1400 Da at 50,000 resolving power with mass accuracies typically within 3 ppm mass error.

Raw data files were processed with BioWorks software (ThermoElectron Corp.) to generate peak lists (DTA). Resulting DTA files from each data acquisition file were searched to identify phosphopeptides against custom databases using the MASCOT search engine (Matrix Science Ltd.; version 2.2.7; in-house licensed). Molecular ions with +1, +2, or +3 charge states determined from an ESI-FTMS primary mass spectrum (LTQ-FT instrument) were typically considered. Searches for trypsin- or chymotrypsin-specific peptide cleavages allowed two missed cleavages, and a mass error tolerance of 5 ppm in the ESI-FT-ICR data and 0.5 Da for MS/MS ions. Peptide modifications included in the searches were oxidation on Met and/or phosphorylation on Ser, Thr, or Tyr residues. Identified peptides were accepted when the MASCOT ion score value exceeded 20, mass errors were less than 5 ppm, and expect values were less than 1. Peptide and phosphopeptide assignments were also manually validated. Sequence coverage of Myc-IST1 was 94%, with identified peptides encompassing 39 out of the 40 Ser and Thr residues. IST1 S4, S99, S153, and S214 phosphorylations were identified in 1, 2, 4, and 1 unique peptides, respectively. Phosphorylation sites were mapped onto the structure of IST1 (PDB ID 3FRR) (*Bajorek et al., 2009b*) using PyMOL (Version 1.3, Schrödinger, LLC, New York, NY).

## Thymidine–nocodazole block

Cells were incubated in 2 mM thymidine DMEM for 18 hr, washed with PBS, released into complete DMEM for 4 hr, and arrested in mitosis using DMEM containing 50 ng/ml nocodazole. Mitotic cells were collected the next day by shake-off, washed in PBS, and lysed in sample buffer. Asynchronous vehicle-treated (DMSO) cultures were lysed at equivalent time points. For siRNA treatments, cell synchronization was initiated 24 hr after the second transfection. Where indicated, the Aurora B inhibitor ZM447439 (Santa Cruz Biotechnology, Dallas, TX) was used at 1 μM final concentration.

## MHC I down-regulation assays

HeLa cells stably expressing KSHV-K3 (KK3) were a kind gift from Prof Paul Lehner (University of Cambridge, UK). HeLa KK3 cells (*Hewitt et al., 2002*) were seeded in a 12-well plate and transfected with siRNA (final concentration 100 nM) as described above. 24 hr after the second siRNA transfection, cells were harvested with PBS 0.5 mM EDTA, washed once in cold PBS, and incubated with PBS containing 2% serum and a FITC-conjugated α-MHC-I antibody (W6/32 clone, AbD Serotec) for 1 hr at 4°C. After washing with PBS 2% serum, cells were resuspended and fixed in 4% paraformaldehyde. Surface MHC-I staining was analyzed by flow cytometry on a FACS-Calibur (Becton–Dickinson, UK).

## Tetherin degradation assay

HT1080/THN-HA and HT1080/THN-HA K5 have been previously described (*Pardieu et al., 2010*). Cells were seeded in 12-well plates and transfected with siRNA (final concentration 100 nM) as described above. 72 hr after the first siRNA transfection, cells were harvested with PBS 0.5 mM EDTA and resuspended in sample buffer. Total Tetherin-HA levels were analyzed by Western blot using α-HA and α-HSP90 antibodies, and visualized using Li-Cor secondary antibodies.

## HIV infectivity

293T cells were transfected with 100 nM of siRNA as described above. 48 hr later, cells were co-transfected with an additional 100 nM of siRNA and 300 ng of HIV pNL/HXB provirus using Lipofectamine 2000. After 48 hr, indicator HeLa-TZM-bl cells (CD4+, CXCR4+, CCR5+, HIV-1 LTR-LacZ) were infected with 100 μl of harvested supernatant from 293T cells. After an additional 48 hr, β-galactosidase activity in cell lysates was measured using the chemiluminescent detection reagent Galacto-Star (Applied Biosystems). In parallel, culture supernatants collected 48 hr after initial transfection were clarified by low-speed centrifugation, and virions were obtained through a 20% sucrose cushion (14,000 rpm, 2 hr). Viral protein contents in cells and particle lysates were analyzed by Western blot using α-Gag antibody.

## Immunofluorescence

Cells were fixed with 4% PFA (20 min) or 0.1% Tween-20, 2% PFA in PBS (10 min), and 100% methanol (2 min). Cells were blocked with PBS 1% BSA or 3% FCS in PBS, stained with the indicated primary antibodies and Alexa 594, 488, or 647 conjugated secondary antibodies (Invitrogen). Nuclei were visualized with Hoechst 33258. Coverslips were mounted in Mowiol. Samples were imaged using an AOBS SP2 confocal microscope (Leica, 60 × 1.4 N.A. oil-immersion objective). The AOTF was used to collect relevant narrow emission-λ windows for each fluorophore. Cells connected by midbodies were identified following tubulin staining, excluding multinucleated cells. For *Figure 4* and *Figure 4—figure supplement 1*, data were collected using an Eclipse Ti-E Inverted CSU-X1 Spinning Disk Confocal (Nikon, Japan) equipped with an Ixon3 EM-CCD camera (Andor, UK). Images were acquired in series of 0.1 μm-spaced Z-stacks with a 100x objective. Deconvolution of 3D stacks was done using *AutoQuant X3* Deconvolution Software (Media Cybernetics, Rockville, MD); projections were obtained using *NIS*-Elements *Ar* Microscope Imaging Software and merged using Photoshop.

## Live imaging

HeLa cells stably expressing YFP/mCherry-Tubulin and/or YFP-LAP2β were seeded on poly-L-lysine coated glass-bottomed 24-well plates (MatTek) and transfected with siRNA as specified. Except where indicated, 24 hr after the second transfection cells were imaged for 24 hr on

a Nikon Ti-Eclipse wide-field inverted microscope (Nikon 40 × 0.75 N.A. dry objective lens) equipped with Perfect Focus system and housed in a 37°C chamber (Solent Scientific, UK) fed with 5% $CO_2$. Multiple fields of view were selected at various XY coordinates, where 3 slices were captured at a 1.25 μm Z-spacing. Images were acquired every 10 min using a CoolSnap HQ2 CCD camera (Photometrics, Tucson, AZ), controlled by NIS-Elements software. Frame-by-frame analysis was performed within NIS-Elements, where abscission time was quantified as the period between midbody formation and severing. Midbody formation was scored as the first frame where two separate cells connected by a compacted bundle of tubulin and fully reformed nuclei were observed. YFP-LAP2β expressing cells were imaged for 72 hr, and chromatin bridge resolution time was scored as the time between nuclear envelope reassembly and bridge resolution. Intercellular bridges that regressed or left the field of view were excluded. For experiments with Aurora B inhibitor, cells at midbody stage were identified prior to treatment with 1 μM ZM447439 (Santa Cruz Biotechnology), when imaging was initiated. Only cells at midbody stage where followed and time to abscission was scored following the same criteria as above. Selected TIF files were exported and assembled in Adobe Photoshop to generate movies with a 200-ms delay between each frame.

## Statistics

All statistical significance was tested using the Mann–Whitney two-tailed U test: ***$p < 0.0001$, **$p = 0.0003$, *$p < 0.05$, ns $p > 0.1$.

## Acknowledgements

We thank the Nikon Imaging Centre at KCL for technical support. We thank the following colleagues for advice or technical assistance: Stuart Neil, Krishna Parsawar (U.U. Mass Spectrometry Core), Scott Endicott (U.U. Peptide Synthesis Core), Chris Hill, John McCullough, Frank Whitby, Leremy Colf, and Kira Miller. University of Utah core facilities were supported in part by Huntsman Cancer Institute Center Support Grant number 5P30CA042014-25. This work was funded by grants from the Medical Research Council (G0802777) and Wellcome Trust (WT102871MA) to JM-S, NIH GM112080 (WIS), NIH GM082545 (SLA and JJS), a KCL School of Medicine PhD Studentship (AC), and an American Cancer Society Postdoctoral Fellowship PF-14-102-01-CSM (DMW). We thank the UK NIHR Comprehensive BRC at KCL for an equipment grant. Use of the Stanford Synchrotron Radiation Lightsource, SLAC National Accelerator Laboratory, is supported by the U.S. Department of Energy, Offices of Science, Office of Basic Energy Sciences under Contract No. DE-AC02-76SF00515. The SSRL Structural Molecular Biology Program is supported by the DOE Office of Biological and Environmental Research, and by the NIH, National Institute of General Medicine Sciences (including P41GM103393). The contents of this publication are solely the responsibility of the authors and do not necessarily represent the official views of NIGMS or NIH. The funding sources were not involved in the study design, data collection and interpretation, or decision to submit the work for publication.

## Additional information

### Competing interests

WIS: Reviewing editor, *eLife*. The other authors declare that no competing interests exist.

### Funding

| Funder | Grant reference | Author |
| --- | --- | --- |
| Medical Research Council (MRC) | G0802777 | Juan Martin-Serrano |
| Wellcome Trust | WT102871MA | Juan Martin-Serrano |
| National Institutes of Health (NIH) | GM112080 | Wesley I Sundquist |
| American Cancer Society | PF-14-102-01-CSM | Dawn M Wenzel |
| University of Utah | Huntsman Cancer Institute (5P30CA042014-25) | Wesley I Sundquist |

| Funder | Grant reference | Author |
|---|---|---|
| Kings College London | PhD Studentship | Anna Caballe |
| U.S. Department of Energy | DE-AC02-76SF00515 | Wesley I Sundquist |
| National Institute of General Medical Sciences (NIGMS) | P41GM103393 | Wesley I Sundquist |
| National Institutes of Health (NIH) | GM082545 | Steven L Alam, Jack J Skalicky |

The funders had no role in study design, data collection and interpretation, or the decision to submit the work for publication.

## Author contributions

AC, Conceived the project, Wrote the manuscript, Performed live-imaging and functional experiments, Performed binding, phosphorylation and microscopy experiments, Conception and design, Acquisition of data, Analysis and interpretation of data, Drafting or revising the article; DMW, Conceived the project, Wrote the manuscript, Performed biochemical, structural and phosphorylation/mass spectrometry studies, Conception and design, Acquisition of data; MA, Conceived the project, Wrote the manuscript, Performed binding, phosphorylation and microscopy experiments, Conception and design, Acquisition of data, Drafting or revising the article; SLA, JJS, Performed structural studies, Acquisition of data, Analysis and interpretation of data; MK, Performed binding experiments, Acquisition of data, Analysis and interpretation of data; JGC, Performed phosphorylation experiments, Acquisition of data, Analysis and interpretation of data; LL, Performed microscopy experiments, Acquisition of data, Analysis and interpretation of data; WIS, Conceived the project, Wrote the manuscript, Conception and design, Analysis and interpretation of data, Drafting or revising the article; JM-S, Conceived the project, Wrote the manuscript, Conception and design, Drafting or revising the article

# Additional files

## Supplementary file

• Supplementary file 1. Supplementary information. Includes list of plasmids used in this study, siRNA sequences, primary antibodies, stable cell lines used in this study.

## Major datasets

The following datasets were generated:

| Author(s) | Year | Dataset title | Dataset ID and/or URL | Database, license, and accessibility information |
|---|---|---|---|---|
| Caballe A, Wenzel DM, Agromayor M, Alam SL, Skalicky JJ, Klock M, Carlton JG, Labrador L, Sundquist WI, Martin-Serrano J | 2014 | ULK3 regulates cytokinetic abscission by phosphorylating ESCRT-III proteins | http://www.rcsb.org/pdb/search/structidSearch.do?structureId=4WZX | Publicly available at RCSB Protein Data Bank (Accession no: 4WZX). |
| Wenzel DM, Skalicky JJ, Sundquist WI | 2014 | Backbone 1H, 13C, and 15N Chemical Shift Assignments for IST1 residues 303-366 | www.bmrb.wisc.edu | Publicly available at Biological Magnetic Resonance Bank (Accession no: 25393). |

The following previously published datasets were used:

| Author(s) | Year | Dataset title | Dataset ID and/or URL | Database, license, and accessibility information |
|---|---|---|---|---|
| Joint Center for Structural Genomics | 2008 | Crystal structure of a phosphoserine aminotransferase serc (chu_0995) from cytophaga hutchinsonii atcc 33406 at 1.75 a resolution | http://www.rcsb.org/pdb/explore/explore.do?structureId=3FFR | Publicly available at RCSB Protein Data Bank (Accession no: 3FFR). |

| Author(s) | Year | Dataset title | Dataset ID and/or URL | Database, license, and accessibility information |
|---|---|---|---|---|
| Stuchell-Brereton MD, Skalicky JJ, Kieffer C, Karren MA, Ghaffarian S, Sundquist WI, | 2007 | VPS4A MIT-CHMP1A complex | http://www.rcsb.org/pdb/explore/explore.do?structureId=2JQ9 | Publicly available at RCSB Protein Data Bank (Accession no: 2JQ9). |
| Kieffer C, Skalicky JJ, Morita E, De Domenico I, Ward DM, Kaplan J, Sundquist WI | 2008 | NMR structure of VPS4A-MIT-CHMP6 | http://www.rcsb.org/pdb/explore/explore.do?structureId=2K3W | Publicly available at RCSB Protein Data Bank (Accession no: 2K3W). |

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
