## [Decision Letter]

Thank you for sending your work entitled "ULK3 regulates cytokinetic abscission by phosphorylating ESCRT-III proteins" for consideration at *eLife*. Your article has been favorably evaluated by Tony Hunter (Senior Editor) and three reviewers, one of whom is a member of our Board of Reviewing Editors.

The Reviewing Editor and the other reviewers discussed their comments before we reached this decision, and the Reviewing Editor has assembled the following comments to help you prepare a revised submission.

The referees’ comments are appended verbatim below. We have also specified the points that need to be addressed (with experiments or text changes).

*Referee 1*:

This is an interesting paper by Martin-Serrano and colleagues on the mechanism of cytokinetic abscission. They identify a protein kinase, ULK3 that binds a number of components of the ESCRT III com-lex. ULK3 is the second kinase (after Aurora B) to enter into this pathway. Abscission happens more rapidly in cells defective in ULK3 and therefore the authors hypothesize that ULK3 is part of the abscission checkpoint that delays abscission until lagging chromosomes are fully segregated and until NPC are properly formed. ULK3 regulates the timing of abscission through phosphorylation and modulation of function of a member of the ESCRT complex protein IST1. ULK3 also functions with CHMP4C to regulate timing of abscission.

1) There is a large body of work in this paper and some important conclusions as well. My main concern stems from how the experiments are done in general, in that the entire study depends on over expression and siRNA based knockdown. In particular all protein-protein interactions have been performed with over expressed ULK3 and localization also done with over expressed ULK3. I think with genome editing working very efficiently through the suite of approaches (ZFN, TALEN, CRISPR/CAS9), I would want to see them make ULK3-HA or ULK3GFP in cell lines expressed off the native promoter to localize the protein and perform immunoprecipitations (for example see the Drubin lab papers on endocytosis in mammalian cells).

It is essential that this point is addressed either with a cell line expressing a chromosomally tagged ULK3 or the use of a ULK3 antibody, since all three referees have raised this point.

2) Similarly, given that ULK3 performs a checkpoint function via IST1 and CHMP4C (i.e. ULK3 is likely non-essential for growth and cell division), they should be also able to make a full deletion of ULK3 in cell lines to determine the precise phenotype. Having a precise deletion will also allow them to address the relationship between ULK3 and Aurora B (parallel vs. linear pathways) and ULK3 and CHMP4C more convincingly. They might also find that ULK3 deletion does not survive, in which case they would have unearthed something new.

You do not have to do this; however, you need to address the issue concerning Aurora B which is pointed out by all three referees.

Thus, despite the fact that the paper is full of very interesting findings, some more precise approaches will fortify the conclusions.

*Referee 2*:

In this manuscript, Caballe et al. report the identification of the kinase ULK3 as a novel regulator of abscission. They show that ULK3 directly binds and phosphorylates several ESCRT-III subunits, and provide a detailed structural characterization of the interactions between the MIT domains of ULK3 and the MIM domains of ESCRT-III subunit IST1. They further show that ULK3 is required to delay abscission in response to the depletion of a nucleoporin (NUP153), to the release of mechanical midbody tension, and to the presence of chromosome bridges at the cleavage plane. The authors further identify ULK3-dependent phosphorylation sites on IST1, which contribute to the regulation of abscission timing, and they show that ULK3 phosphorylates CHMP4C to mediate the abscission delay. Based on these data, the authors propose that ULK3 is a regulator of a previously identified "abscission checkpoint" that monitors chromosome segregation and nuclear pore integrity. Overall, the manuscript presents high quality data and convincing interpretation. The new insights into the regulation of abscission are very interesting and relevant for a broad cell biological readership.

Specific comments:

1) The authors show that depletion of ULK3 leads to premature resolution of anaphase chromosome bridges and use this as an argument that ULK3 functions in a previously identified Aurora B-dependent "abscission checkpoint" (Steigemann et al., 2009). The previous study, however, reported that loss-of-function of the Aurora B-dependent "abscission checkpoint" results in cleavage furrow regression rather than in premature resolution of chromosome bridges. The authors explain the different phenotypes by proposing that ULK3 functions downstream of Aurora B. There are, however, no data directly supporting this interpretation and ULK3 could also have a function independent of Aurora B. The authors should rephrase the Discussion accordingly.

It is essential you do this, since all three referees make this point.

2) The authors show that overexpressed HA-ULK3 co-localizes with IST1, either at the Flemming body (Figure 4) or at midbody rings (Figure 4—figure supplement 1). As the overexpression of HA-ULK3 may potentially result in its mislocalization, it would be important to validate this observation by staining with antibodies against endogenous ULK3 in unperturbed conditions. In addition, quantification and statistical data analysis should be added to the ULK3 localization experiments shown in Figure 4 and Figure 4—figure supplement 1.

It is essential you address this, since all three referees have asked for this.

*Referee 3*:

In this study, Caballe et al., describe the role of a novel protein kinase, ULK3, in the regulation of cytokinetic abscission. Using an impressive range of experimental approaches, this study characterizes ULK3 and dissects its interplay with ESCRT-III proteins as key regulators of abscission. An especially tight interaction was found between ULK3 and the ESCRT-III component IST1, and the authors determined the crystal structure of the ULK3 MIT2-IST1 MIM1 complex at 1.4 Å resolution. Importantly, by invoking a novel kinase as a central regulatory node, this study substantially develops our understanding of the poorly characterized abscission checkpoint. Considering the importance of cytokinetic/abscission fidelity in maintenance of genome integrity and prevention of tumorigenesis, this manuscript is of potential interest to a broad readership.

However, even though this study brings forward numerous exciting observations regarding ULK3 and ESCRT-III proteins, it does not always manage to combine these observations into a coherent story with a clear-cut molecular mechanism of action. Furthermore, the manuscript fails to relate ULK3 function to existing literature describing the abscission checkpoint and implicating Aurora B and downstream components.

Specific comments:

1) The authors state that ULK3 is part of the Aurora B-mediated abscission checkpoint. If they want to maintain this statement they should further explore the ULK3-Aurora B relationship. Using similar nocodazole-release experiments to those shown in Figure 7—figure supplement 1, the authors have previously concluded that the mobility shift of CHMP4C during mitosis (due to phosphorylation) was dependent on Aurora B activity (PMID:22422861; Figure 4). In turn, CHMP4C phosphorylation by Aurora B was found to depend on the 'INS' region. Using the same setup in the current study, the authors find that deleting the 'INS' region has no effect on CHMP4C phosphorylation during mitosis, with ULK3 being the main regulator of mitotic phosphorylation in the current study (Figure 7). How do the authors explain this apparent discrepancy? To this end, and to mechanistically link the current study with existing literature, the authors should explore the relationship between Aurora B and ULK3 in more detail. Does ULK3 control interaction of Aurora B with CHMP4C? Does Aurora B provide the additional phosphorylation on IST1? This can be addressed using Aurora B inhibitors.

This point needs to be addressed with experiments as all referees have raised this.

2) The live imaging described in this study was performed by taking 3 z sections at 1.25μM intervals, and imaging every 10 minutes. This appears very little when considering the mildness of ULK3 phenotypes (with the mean difference sometimes as little as 20 minutes), the apparent variation between experiments (NT control mean values ranging from 80 to 100 minutes) and the fact that intercellular bridges can move significantly in Z, thus temporarily moving out of the plane of focus and obscuring time of abscission. Another consequence is poor temporal alignment of cells as exemplified in Figure 2—figure supplement 1 (compare the t=0 time point for the non-targeting vs. ULK3 siRNAs in panel D; these cells are very different). To give the reader a better indication of the phenotypes involved, mean abscission time and associated standard deviation for important phenotypes should be mentioned in the Results text, as should the imaging frequency.

This point is important and needs to be validated with experiments.

3) The data relating to cell density (Figure 2) are very interesting as they hint towards a role for ULK3 in tension-dependent abscission control, but they complicate interpretations of the other live imaging panels. These tension effects are larger (120min vs. 70min) than most other ULK3 live imaging phenotypes in Figures 2 and 3 (with only Figure 3 exceeding this). The authors need to describe whether and how cell density, cell mobility and other tensile variables are considered and excluded as possible contributors to the phenotypes described in the other figure panels.

This requires careful explanation, but experiments are not required.

4) ULK3-dependent regulation of ESCRT-III subunits by phosphorylation is potentially one of the most exciting points of the study. However, in the current presentation, this regulation is incoherent and lacks mechanistic depth. For example, the authors describe ULK3 dependent phosphorylation of IST1, and identify 4 phosphorylation sites in IST1. However, it is unclear which of these are attributed to ULK3 and which are involved in abscission checkpoint signaling, complicating interpretation of results with the 4SA/E mutants in relation to ULK3. The authors should assess and clarify whether all 4 sites are ULK3 dependent, and if not, which ones are. This analysis could be done using in vitro kinase assays on recombinant proteins or by purifying IST1 from ULK3 depleted cells, followed by phospho-MS/MS analysis. Have the authors analyzed the single SA mutants? This is important since IST1 phospho-mutant phenotypes do not mimic ULK3 phenotypes, with ULK3 overexpression only mildly delaying abscission (Figure 3), but the IST1 4SE showing strong effects on abscission and multinucleation (Figure 6). This discrepancy should be addressed.

Furthermore, does the IST1 4SA mutant mimic ULK3 depletion with respect to tension control of abscission or abscission in presence of chromatin bridges? Conversely, can the IST1 4SE mutant sustain checkpoint arrest in the absence of ULK3? (This is essential).

5) Overexpression of ULK3 alters the localization frequencies of IST1 and CHMP4B to the Flemming body. The authors should complement with localization studies in ULK3-depleted cells to rule out overexpression-induced artifacts.

This is essential.

6) A very interesting lead is provided by evidence that phosphorylated IST1 shows remarkably increased interaction with VPS4 and LIP5, but the implications of this exciting observation are not explored in a cellular context. This should be done, especially when considering the central position of VPS4 in the abscission checkpoint and ESCRT-III dynamics in general. Is VPS4 localization affected in the IST1 4SE mutant, and where does the 4SE mutant itself localize in normal cells? Does it localize to the Flemming body, as could be expected from Figure 4? Furthermore, the position of the IST1 phosphorylation sites outside the MIM motifs leads the authors to suggest the existence of a bridging factor, but to lend more credibility to this statement the IST1 4SE mutant should be included in the yeast-2-hybrid setup shown in Figure 6—figure supplement 1. Considering the published interaction between ANCHR, CHMP4C and VPS4, the authors should explore whether IST1 4SE, ANCHR and CHMP4C associate, and if so, whether this is stronger than for the WT IST1.

You do not have to do this, since it is beyond the scope of a study on IST1.

7) Furthermore, the relation between ULK3 and other ESCRT-III subunits should be explored further to see whether there is a more general trend in ULK3 effects on ESCRT-III: Does ULK3 overexpression similarly relocalize other phosphorylation targets such as CHMP1A, CHMP1B and CHMP2A to the Flemming body?

It is not essential that you do this, since the paper is mainly about IST1.

8) Figure 3 provide a critical argument that actual kinase activity is required for ULK3 function during abscission. However, Figure 3 suggests that association of kinase dead ULK3 (K44H) with ESCRT-III subunits is more tight than the WT ULK3 (2-10 fold higher beta-galactosidase activity), which could have major effects on cellular function of ULK3 and ESCRT-III. Since Figure 1 suggests that kinase activity does not affect IST1 interaction with ULK3, is it possible that K44H affects ULK3 structure beyond its effect on kinase activity? This should be discussed. Also, since IST1 is a major component of this study, Figure 3 should include IST1 interaction data.

This is mostly about rewriting with some data that you might already have.

9) The proposed ULK3-CHMP4C link is not entirely convincing and seems almost like an afterthought. These proteins do not interact, and the authors do not explore the significance of the phosphorylation event in any detail (e.g. no mapping of target sites, no ULK3-specific phosphoCHMP4C mutants to see if this affects abscission timing etc.). On the other hand, several datasets points towards interplay between CHMP4C and ULK3 that are left undiscussed: For example, ULK3 KD seems to affect the overall level of CHMP4C (Figure 2). Can the authors explain this? Does ULK3 control CHMP4C stability, and can this be rescued by WT, K44M or M434D ULK3 alleles? Additionally, ULK3 SDS-PAGE mobility is altered upon CHMP4C knockdown (Figure 5). Could this reflect altered ULK3 modifications, such as phosphorylation? This should be discussed. In the absence of further data the authors should play down their statements about the ULK3-CHMP4C link (This part could be deleted).

---

## [Author Response]

Referee 1:

1) There is a large body of work in this paper and some important conclusions as well. My main concern stems from how the experiments are done in general, in that the entire study depends on over expression and siRNA based knock down. In particular all protein-protein interactions have been performed with over expressed ULK3 and localization also done with over expressed ULK3. I think with genome editing working very efficiently through the suite of approaches (ZFN, TALEN, CRISPR/CAS9), I would want to see them make ULK3-HA or ULK3GFP in cell lines expressed off the native promoter to localize the protein and perform immunoprecipitations (for example see the Drubin lab papers on endocytosis in mammalian cells).

*It is essential that this point is addressed either with a cell line expressing a chromosomally tagged ULK3 or the use of a ULK3 antibody, since all three referees have raised this point*.

Regarding the reviewer’s concern with our experimental strategy of using siRNA followed by rescue with siRNA-resistant constructs (Figure 3), we would emphasize that this approach remains a robust experimental approach in which functional studies can exclude off-target effects. Nonetheless, we have taken the reviewer’s advice and used the CRISPR-Cas9 system to delete the ULK3 gene. New data in Figure 2 shows that full deletion of ULK3 in two separate cell clones results in abscission checkpoint failure in response to disrupted nucleopores. This result is entirely consistent with our siRNA data.

We have also added new data to strengthen the conclusion that ULK3 and IST1 interact at physiological levels. In particular we have fused ULK3 to the R113G mutant of the *E. coli* BirA biotin ligase, which biotinylates lysine residues of proteins in close proximity. The BirA-ULK3 construct was then stably expressed in HeLa cells that were supplemented with biotin and subjected to streptavidin precipitation, demonstrating the specific association of BirA-ULK3 with the endogenous IST1 (Figure 1). This new data is consistent with our result in Figure 1, showing the interaction of HA-IST1 with the endogenous ULK3.

Lastly, we have now identified a commercial antibody that detects ULK3 in immunofluorescence imaging experiments. The new data in Figure 2 using this antibody shows the localization of endogenous ULK3 at the central region of the midbody. We also show in Figure 7 that endogenous ULK3 co-localizes at the midbody with GFP-CHMP4C expressed at physiological levels, thus further supporting a role for ULK3 in cytokinetic abscission and a functional association between ULK3 and CHMP4C.

*2) Similarly, given that ULK3 performs a checkpoint function via IST1 and CHMP4C (i.e. ULK3 is likely non-essential for growth and cell division), they should be also able to make a full deletion of ULK3 in cell lines to determine the precise phenotype. Having a precise deletion will also allow them to address the relationship between ULK3 and Aurora B (parallel vs. linear pathways) and ULK3 and CHMP4C more convincingly. They might also find that ULK3 deletion does not survive, in which case they would have unearthed something new*.

*You do not have to do this; however, you need to address the issue concerning Aurora B which is pointed out by all three referees*.

As explained above, our new data show that cell lines deleted of ULK3 survive, but they do not sustain the abscission checkpoint activity. The fact that these cells are viable is consistent with our model that ULK3’s main function is to regulate the abscission machinery.

As requested by the three reviewers, we have addressed further the functional relationship between ULK3 and Aurora B and we now provide compelling evidence that abscission regulation by ULK3 requires Aurora B. More specifically, we show that the Aurora B inhibitor ZM447439 abrogates the abscission delays induced by ULK3 overexpression (Figure 7). We also show that the treatment with the Aurora B inhibitor induces alterations in the mitotic phosphorylation of IST1 (Figure 7), a phenotype that is reminiscent of the effects in IST1 phosphorylation observed in ULK3-depleted cells (Figure 5). Lastly, we have further analysed midbodies in the context of nucleopore disruption, showing that active Aurora B persists at the midbody in ULK3-depleted cells (Figure 2—figure supplement 1). Altogether, these new lines of evidence support our original model whereby ULK3 is a downstream component of the Aurora B-dependent abscission checkpoint.

Referee 2:

*In this manuscript, Caballe et al. report the identification of the kinase ULK3 as a novel regulator of abscission. They show that ULK3 directly binds and phosphorylates several ESCRT-III subunits, and provide a detailed structural characterization of the interactions between the MIT domains of ULK3 and the MIM domains of ESCRT-III subunit IST1. They further show that ULK3 is required to delay abscission in response to the depletion of a nucleoporin (NUP153), to the release of mechanical midbody tension, and to the presence of chromosome bridges at the cleavage plane. The authors further identify ULK3-dependent phosphorylation sites on IST1, which contribute to the regulation of abscission timing, and they show that ULK3 phosphorylates CHMP4C to mediate the abscission delay. Based on these data, the authors propose that ULK3 is a regulator of a previously identified "abscission checkpoint" that monitors chromosome segregation and nuclear pore integrity. Overall, the manuscript presents high quality data and convincing interpretation. The new insights into the regulation of abscission are very interesting and relevant for a broad cell biological readership*.

*Specific comments*:

*1) The authors show that depletion of ULK3 leads to premature resolution of anaphase chromosome bridges and use this as an argument that ULK3 functions in a previously identified Aurora B-dependent "abscission checkpoint" (Steigemann et al., 2009). The previous study, however, reported that loss-of-function of the Aurora B-dependent "abscission checkpoint" results in cleavage furrow regression rather than in premature resolution of chromosome bridges. The authors explain the different phenotypes by proposing that ULK3 functions downstream of Aurora B. There are, however, no data directly supporting this interpretation and ULK3 could also have a function independent of Aurora B. The authors should rephrase the Discussion accordingly*.

*It is essential you do this, since all three referees make this point*.

As discussed above we provide three new pieces of information supporting the idea that ULK3 functions downstream of Aurora B in the regulation of abscission.

*2) The authors show that overexpressed HA-ULK3 co-localizes with IST1, either at the Flemming body (*Figure 4*) or at midbody rings (*Figure 4—figure supplement 1*). As the overexpression of HA-ULK3 may potentially result in its mislocalization, it would be important to validate this observation by staining with antibodies against endogenous ULK3 in unperturbed conditions. In addition, quantification and statistical data analysis should be added to the ULK3 localization experiments shown in*
Figure 4
*and*
Figure 4—figure supplement 1.

*It is essential you address this, since all three referees have asked for this*.

As noted above, we have now shown that endogenous ULK3 localizes to the central region of the midbody. Nevertheless, we thank the reviewer for pointing the lack of clarity in our description of the localization of overexpressed HA-ULK3. This construct localizes to both the Flemming body (42% of midbodies analyzed) and the midbody rings (58%). We now have added a panel in both Figure 4 and Figure 4—figure supplement 1 to show these localizations and we have included the corresponding quantification data in the figure legend. We have also scored the localization of the endogenous ULK3, finding a Flemming body localization in 93% of the cases vs 7% of cases showing localization to the midbody rings. Therefore overexpressed HA-ULK3 is not mislocalized although it has a higher tendency to localize to the midbody rings. Our interpretation of this difference is that the endogenous ULK3 likely shuttles between the Flemming body and the rings, whereas the overexpressed protein is more stabilized at the rings.

Owing to technical limitations, the co-localization of endogenous ULK3 and IST1 cannot be determined because both antibodies were raised in rabbits.

Referee 3:

*Even though this study brings forward numerous exciting observations regarding ULK3 and ESCRT-III proteins, it does not always manage to combine these observations into a coherent story with a clear-cut molecular mechanism of action. Furthermore, the manuscript fails to relate ULK3 function to existing literature describing the abscission checkpoint and implicating Aurora B and downstream components*.

*Specific comments*:

*1) The authors state that ULK3 is part of the Aurora B-mediated abscission checkpoint. If they want to maintain this statement they should further explore the ULK3-Aurora B relationship. Using similar nocodazole-release experiments to those shown in Figure7–figure supplement 1, the authors have previously concluded that the mobility shift of CHMP4C during mitosis (due to phosphorylation) was dependent on Aurora B activity (PMID:22422861;*
Figure 4*). In turn, CHMP4C phosphorylation by Aurora B was found to depend on the 'INS' region. Using the same setup in the current study, the authors find that deleting the 'INS' region has no effect on CHMP4C phosphorylation during mitosis, with ULK3 being the main regulator of mitotic phosphorylation in the current study (*Figure 7*). How do the authors explain this apparent discrepancy? To this end, and to mechanistically link the current study with existing literature, the authors should explore the relationship between Aurora B and ULK3 in more detail. Does ULK3 control interaction of Aurora B with CHMP4C? Does Aurora B provide the additional phosphorylation on IST1? This can be addressed using Aurora B inhibitors*.

*This point needs to be addressed with experiments as all referees have raised this*.

As discussed above, our new data demonstrate a functional connection between ULK3 and Aurora B, and support the notion that ULK3 is a downstream effector of Aurora B in the abscission checkpoint. This connection is further supported by our observation that mitotic phosphorylation of IST1 is reduced in cells treated with Aurora B inhibitor. This experiment shows that this additional phosphorylation of IST1 requires Aurora B but does not show that Aurora B itself phosphorylates IST1. We believe that a more likely alternative is that Aurora B activates a downstream kinase, such as ULK3, that phosphorylates IST1.

Based on the reviewer’s comments we realized that further clarification is needed to answer the apparent discrepancy on the Aurora B requirement for CHMP4C phosphorylation. It should be pointed out that we have looked at CHMP4C phosphorylation with two different approaches, by looking at changes in mobility using SDS-PAGE and by performing in vitro phosphorylation assays with recombinant kinases. These methods complement each other although it is well known that phosphorylation events do not result necessarily in mobility shifts. Our published data and the results in this manuscript indicate that Aurora B and ULK3 phosphorylate CHMP4C at separate sites with different consequences on the mobility of this protein in SDS-PAGE. In this context, we previously observed that the INS region is required for in vitro phosphorylation of CHMP4C by recombinant Aurora B (PMID:22422861; Figure 4), but we never showed that this insertion is required for the mobility shift observed in mitotically arrested cells. We have now done these experiments showing that deletion of the INS region does not alter the mobility shift of CHMP4C in mitosis whereas ULK3 depletion inhibits this shift (Figure 7). The fact that Aurora B inhibitors reduce the mitotic phosphorylation of CHMP4C as measured by mobility shifts (PMID:22422861; Figure 4) is now better explained by the new data indicating that ULK3 is a downstream effector of Aurora B, thus suggesting that Aurora B inhibition is likely to have a knock-on effect on ULK3 activity thus preventing the mitotic phosphorylation of CHMP4C. We have now included a model as the striking image to clarify this point.

Lastly, we agree with the reviewer that regulation of the Aurora B/CHMP4C interaction by ULK3 is possible. However, the Aurora B/CHMP4C interaction is not detectable by conventional methods other than in vitro kinase assays. This is not entirely surprising as the interaction of Aurora B with its physiological substrates is thought to be transient and these physical interactions have been rarely reported in the literature.

*2) The live imaging described in this study was performed by taking 3 z sections at 1.25μM intervals, and imaging every 10 minutes. This appears very little when considering the mildness of ULK3 phenotypes (with the mean difference sometimes as little as 20 minutes), the apparent variation between experiments (NT control mean values ranging from 80 to 100 minutes) and the fact that intercellular bridges can move significantly in Z, thus temporarily moving out of the plane of focus and obscuring time of abscission. Another consequence is poor temporal alignment of cells as exemplified in*
Figure 2—figure supplement 1
*(compare the t=0 time point for the non-targeting vs. ULK3 siRNAs in panel D; these cells are very different). To give the reader a better indication of the phenotypes involved, mean abscission time and associated standard deviation for important phenotypes should be mentioned in the Results text, as should the imaging frequency*.

*This point is important and needs to be validated with experiments*.

We disagree with the reviewer regarding the “mildness” of the ULK3 phenotypes. We emphasize the fact that these differences are highly significant and the magnitude of this difference is equivalent to the effects observed in CHMP4C-depleted cells, thus highlighting the biological relevance of this phenotype. Regarding the apparent variation between experiments, this reflects the normal day-to-day variation in abscission time that is observed in HeLa cells and our experiments are specifically designed to neutralize this issue. Crucially, equivalent cell cultures were treated in parallel with the different siRNAs, thus excluding cell density as a source of uncontrolled variability. Moreover, the data were acquired in parallel using a motorized microscope to ensure that the differences in abscission time are meaningful. Importantly, the small fraction of midbodies that moved out of the plane of focus were excluded from the analysis. Regarding the temporal alignment of cells, it is worth noting that midbody formation occurs less than 10 minutes after the end of anaphase. Hence, the apparent differences pointed by the reviewer represent stages that are very close in time. Moreover, to ensure that the differences observed are not due to potential artifacts associated with the scoring of midbody formation, we have re-analyzed an entire figure (including three independent experiments) using the end of anaphase instead of midbody formation as t=0. As Figure 8 shows, both types of analysis result in similar differences in the abscission time. The slightly longer abscission times observed when end of anaphase was used as t=0 are expected as this is an earlier event than midbody formation.

The mean abscission time and standard deviations are now mentioned in the text.

Author response image 1.**DOI:**
http://dx.doi.org/10.7554/eLife.06547.049

*3) The data relating to cell density (*Figure 2*) are very interesting as they hint towards a role for ULK3 in tension-dependent abscission control, but they complicate interpretations of the other live imaging panels. These tension effects are larger (120min vs. 70min) than most other ULK3 live imaging phenotypes in*
Figures 2 and 3
*(with only*
Figure 3
*exceeding this). The authors need to describe whether and how cell density, cell mobility and other tensile variables are considered and excluded as possible contributors to the phenotypes described in the other figure panels*.

*This requires careful explanation, but experiments are not required*.

Cell division events included in the tension analysis represent extreme cases of differing cell densities within the experiment, thus explaining the larger differences in abscission time observed upon ULK3 depletion. We now have modified the text to make this point clearer.

*4) ULK3-dependent regulation of ESCRT-III subunits by phosphorylation is potentially one of the most exciting points of the study. However, in the current presentation, this regulation is incoherent and lacks mechanistic depth. For example, the authors describe ULK3 dependent phosphorylation of IST1, and identify 4 phosphorylation sites in IST1. However, it is unclear which of these are attributed to ULK3 and which are involved in abscission checkpoint signaling, complicating interpretation of results with the 4SA/E mutants in relation to ULK3. The authors should assess and clarify whether all 4 sites are ULK3 dependent, and if not, which ones are. This analysis could be done using in vitro kinase assays on recombinant proteins or by purifying IST1 from ULK3 depleted cells, followed by phospho-MS/MS analysis. Have the authors analyzed the single SA mutants? This is important since IST1 phospho-mutant phenotypes do not mimic ULK3 phenotypes, with ULK3 overexpression only mildly delaying abscission (*Figure 3*), but the IST1 4SE showing strong effects on abscission and multinucleation (*Figure 6*). This discrepancy should be addressed*.

*Furthermore, does the IST1 4SA mutant mimic ULK3 depletion with respect to tension control of abscission or abscission in presence of chromatin bridges? Conversely, can the IST1 4SE mutant sustain checkpoint arrest in the absence of ULK3? (This is essential)*.

A) Regarding the reviewers’ concern that the 4SE mutant phenotype is more severe than our ULK3 overexpression phenotype, we would argue that this is the expected result because, the phosphomimetic Ser to Glu mutations are irreversible. We envision a model in which ULK3 delays abscission by phosphorylating IST1, rendering it incapable of functioning in abscission. If cells do not contain defects like lagging chromatin or misformed nuclear pores, they are released from the checkpoint, IST1 is dephosphorylated, and abscission proceeds. This model is consistent with our observation that IST1 cycles between phosphorylated and de-phosphorylated states throughout the cell cycle, with the phospho-IST1 species accumulating during mitosis (as mimicked by nocodazole treatment, Figure 5 and Figure 7). This “dephosphorylation” event cannot occur for the 4SE IST1 mutant. Moreover, we would point out that cells that express the IST1 4SE that escape abscission failure have delays (48 min) (Figure 6), that are comparable to the delays induced by ULK3 overexpression (78 min and 39 min in two different experiments). Thus, IST1 4SE either inhibits abscission altogether or induces abscission delays that are comparable to ULK3 overexpression.

B) In response to the reviewer’s question whether the IST1 4SA mutant mimics ULK3 depletion with respect to tension control of abscission or abscission in presence of chromatin bridges, we have now examined the activities of the IST1 4SA mutant under both of these conditions (in addition to nuclear pore depletion, which was included in our original submission (Figure 6, Figure 6—figure supplement 1 and Figure 6, respectively). As expected the IST1 4SA mutant fails to delay abscission in response to nuclear pore depletion and lagging chromatin at the intercellular bridge. Surprisingly, however, the IST1 4SA mutant behaves like the wild type IST1 protein in response to tension. These observations imply that abscission delays that are triggered via (lack of) tension require different signals and/or may not be as sensitive to IST1 phosphorylation as delays triggered through nuclear pores or lagging chromatin. This result is interesting because it could mean that delays in abscission in response to tension may require additional ULK3 substrates and/or additional, unidentified phosphorylation sites on IST1.

*5) Overexpression of ULK3 alters the localization frequencies of IST1 and CHMP4B to the Flemming body. The authors should complement with localization studies in ULK3-depleted cells to rule out overexpression-induced artifacts*.

*This is essential*.

We appreciate the reviewer’s suggestion. However, we believe that the feasibility of the proposed experiment is unclear. Only a very small fraction of abscission events will be significantly delayed by the abscission checkpoint in unperturbed situations. For example, less than 5% of the cell division events have lagging chromosomes that delay abscission. Thus, the analysis of CHMP4B/IST1 localization in ULK3-depleted cells would mainly reveal the localization of these proteins in abscission events in which the abscission checkpoint would not normally be on. In contrast, ULK3 overexpression provides a unique opportunity for meaningful localization studies because this situation induces (or mimics) the abscission checkpoint in a large fraction of the cell division events. Additionally, our claim of ULK3-dependent midbody CHMP4B and IST1 relocalization is further supported by our new data showing that endogenous ULK3 and overexpressed ULK3 exhibit similar localization patterns.

*6) A very interesting lead is provided by evidence that phosphorylated IST1 shows remarkably increased interaction with VPS4 and LIP5, but the implications of this exciting observation are not explored in a cellular context. This should be done, especially when considering the central position of VPS4 in the abscission checkpoint and ESCRT-III dynamics in general. Is VPS4 localization affected in the IST1 4SE mutant, and where does the 4SE mutant itself localize in normal cells? Does it localize to the Flemming body, as could be expected from*
Figure 4*? Furthermore, the position of the IST1 phosphorylation sites outside the MIM motifs leads the authors to suggest the existence of a bridging factor, but to lend more credibility to this statement the IST1 4SE mutant should be included in the yeast-2-hybrid setup shown in*
Figure 6—figure supplement 1*. Considering the published interaction between ANCHR, CHMP4C and VPS4, the authors should explore whether IST1 4SE, ANCHR and CHMP4C associate, and if so, whether this is stronger than for the WT IST1*.

*You do not have to do this, since it is beyond the scope of a study on IST1*.

*7) Furthermore, the relation between ULK3 and other ESCRT-III subunits should be explored further to see whether there is a more general trend in ULK3 effects on ESCRT-III: Does ULK3 overexpression similarly relocalize other phosphorylation targets such as CHMP1A, CHMP1B and CHMP2A to the Flemming body*?

*It is not essential that you do this, since the paper is mainly about IST1*.

*8)*
Figure 3
*provide a critical argument that actual kinase activity is required for ULK3 function during abscission. However,*
Figure 3
*suggests that association of kinase dead ULK3 (K44H) with ESCRT-III subunits is more tight than the WT ULK3 (2-10 fold higher beta-galactosidase activity), which could have major effects on cellular function of ULK3 and ESCRT-III. Since*
Figure 1
*suggests that kinase activity does not affect IST1 interaction with ULK3, is it possible that K44H affects ULK3 structure beyond its effect on kinase activity? This should be discussed. Also, since IST1 is a major component of this study,*
Figure 3
*should include IST1 interaction data*.

*This is mostly about rewriting with some data that you might already have*.

We agree with the reviewer that the K44H mutant gives a stronger signal than WT ULK3 with the ESCRT-III subunits but these differences should be interpreted with caution as the quantitative value of the yeast two-hybrid assay is very limited (beyond revealing whether or not two proteins interact). Moreover, the binding of WT or K44H ULK3 to ESCRT-III subunits such as IST1 is comparable when these interactions are tested by co-precipitation assays (see Figure 9), suggesting that the K44H mutation does not affect the ULK3 structure beyond its kinase activity.

Author response image 2.**DOI:**
http://dx.doi.org/10.7554/eLife.06547.050

As noted in the text, two hybrid data for the ULK3/IST1 interaction are not included in Figure 3 (or elsewhere) because both proteins induce reporter gene expression when they are fused to the GAL4 DNA binding domain.

*9) The proposed ULK3-CHMP4C link is not entirely convincing and seems almost like an afterthought. These proteins do not interact, and the authors do not explore the significance of the phosphorylation event in any detail (e.g. no mapping of target sites, no ULK3-specific phosphoCHMP4C mutants to see if this affects abscission timing etc.). On the other hand, several datasets points towards interplay between CHMP4C and ULK3 that are left undiscussed: For example, ULK3 KD seems to affect the overall level of CHMP4C (*Figure 2*). Can the authors explain this? Does ULK3 control CHMP4C stability, and can this be rescued by WT, K44M or M434D ULK3 alleles? Additionally, ULK3 SDS-PAGE mobility is altered upon CHMP4C knockdown (*Figure 5*). Could this reflect altered ULK3 modifications, such as phosphorylation? This should be discussed. In the absence of further data the authors should play down their statements about the ULK3-CHMP4C link (This part could be deleted)*.

We agree that this is not a major aspect of our study, but we nevertheless believe that our paper is strengthened by its inclusion.